# 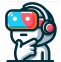 EOC-Bench: Can MLLMs Identify, Recall, and Forecast Objects in an Egocentric World?

Yuqian Yuan[1,2,3*] Ronghao Dang[2,3*] Long Li[2,3*] Wentong Li[1*] Dian Jiao[1]
Xin Li[2,3] Deli Zhao[2,3] Fan Wang[2,3] Wenqiao Zhang[1†] Jun Xiao[1] Yueting Zhuang[1]
[1]Zhejiang University  [2]DAMO Academy, Alibaba Group  [3]Hupan Lab

🌐 Project Page      ⚙ Code      🤗 Benchmark

## Abstract

The emergence of multimodal large language models (MLLMs) has driven break-throughs in egocentric vision applications. These applications necessitate persistent, context-aware understanding of objects, as users interact with tools in dynamic and cluttered environments. However, existing embodied benchmarks primarily focus on static scene exploration, emphasizing object's appearance and spatial attributes while neglecting the assessment of dynamic changes arising from users' interactions. To address this gap, we introduce `EOC-Bench`, an innovative benchmark designed to systematically evaluate object-centric embodied cognition in dynamic egocentric scenarios. Specially, `EOC-Bench` features 3,277 meticulously annotated QA pairs categorized into three temporal categories: Past, Present, and Future, covering 11 fine-grained evaluation dimensions and 3 visual object referencing types. To ensure thorough assessment, we develop a mixed-format human-in-the-loop annotation framework with four types of questions and design a novel multi-scale temporal accuracy metric for open-ended temporal evaluation. Based on `EOC-Bench`, we conduct comprehensive evaluations of various proprietary, open-source, and object-level MLLMs. `EOC-Bench` serves as a crucial tool for advancing the embodied object cognitive capabilities of MLLMs, establishing a robust foundation for developing reliable core models for embodied systems.

## 1 Introduction

The rapid advancement of multimodal large language models (MLLMs) [1, 2, 3, 4] has paved the way for the development of intelligent systems that can comprehend and interact with the visual world. Among these innovations, egocentric vision, where systems perceive environments from a human-like first-person perspective, has gained significant attention due to its critical applications in fields such as augmented reality [5], embodied AI [6, 7] and robotic manipulation [8, 9, 10].

Understanding objects precisely within egocentric contexts presents unique challenges that extend beyond conventional vision tasks. It demands a continuously evolving, context-aware comprehension of objects, encompassing their types, usages, states, and interactions, as users dynamically interact with tools and undertake various operational tasks. In egocentric environments, particularly in densely cluttered settings like kitchens and laboratories, objects exhibit three critical properties: (1) **Fleeting visibility**, indicating dynamic changes in state and position due to frequent occlusions and shifts in viewpoint; (2) **Visual ambiguity**, arising from similar-looking items in close spatial proximity; and

---

*Equal contribution.
†Corresponding author.

39th Conference on Neural Information Processing Systems (NeurIPS 2025) Track on Datasets and Benchmarks.

| Benchmark | #Videos | #Samples | Question Type | Annotator | Real Scenes | Egocentric | Object Dynamics | Time Sensitive | Visual Prompts |
|---|---|---|---|---|---|---|---|---|---|
| MVBench [11] | 3,673 | 4,000 | Close | Template | ✓ | ✗ | ✓ | ✗ | ✗ |
| VideoRefer-Bench [12] | 598 | 1,400 | Open/Close | Human | ✓ | ✗ | ✓ | ✗ | $\mathcal{M}$ |
| Charades-STA [13] | 1,334 | 4,233 | Open | Automatic/Human | ✓ | ✗ | ✓ | ✓ | ✗ |
| ScanQA [14] | - | 4,976 | Open | Automatic/Human | ✓ | ✓ | ✗ | ✗ | ✗ |
| SQA3D [15] | - | 2,143 | Open | Human | ✓ | ✓ | ✗ | ✗ | $\mathcal{A}$ |
| Env-QA [16] | 3,489 | 12,760 | Open | Template | ✗ | ✓ | ✓ | ✗ | ✗ |
| OpenEQA [17] | 180 | 1,600 | Open | Human | ✓ | ✓ | ✗ | ✗ | ✗ |
| VSI-Bench [18] | 288 | 5,000 | Open/Close | Template/Human | ✓ | ✓ | ✗ | ✗ | ✗ |
| ECBench [19] | 386 | 4,324 | Open/Close | Human | ✓ | ✓ | ✓(6%) | ✗ | ✗ |
| EOC-Bench (**Ours**) | 656 | 3,277 | Open/Close | Human | ✓ | ✓ | ✓ | ✓ | $\mathcal{P}, \mathcal{B}, \mathcal{M}$ |

Table 1: Comparison of widely adopted Embodied/General VideoQA benchmarks with our EOC-Bench. $\mathcal{P}, \mathcal{B}, \mathcal{M}$ and $\mathcal{A}$ represent visual prompts for object referencing, specifically as point, box, mask and arrow, respectively.

(3) **Temporal dependency**, where current states rely on historical interactions and inform future outcomes. Successful object perception in these scenarios requires models capable of sustaining persistent visual grounding while simultaneously processing spatiotemporal details. Unfortunately, existing benchmarks fail to systematically evaluate this capability.

As shown in Table 1, existing embodied benchmarks, such as the closed-vocabulary ScanQA [14] and SQA3D [15], focus on understanding static scene through closed-vocabulary queries based on task-specific datasets. Consequently, these benchmarks lack the scope to evaluate task-generalized MLLMs for broader cognitive capabilities. More recently, OpenEQA [17], VSI-Bench [18], and ECBench [19] have developed open-vocabulary benchmarks to evaluate MLLMs' question-answering (QA) capabilities in indoor embodied video contexts. Despite these promising advancements, current benchmarks primarily concentrate on static scene exploration, such as home tours, and predominantly evaluate appearance and spatial attributes. They often overlook dynamic interactions within egocentric operational environments, where users engage actively with tools and perform various tasks involving objects. Building these capabilities is crucial for advancing embodied system.

To bridge this gap, we introduce EOC-Bench, a novel object-centric video benchmark that rigorously evaluates MLLMs's object cognition capabilities in egocentric operational scenarios. Built on the idea that effective AI assistants must comprehensively comprehend objects across temporal dimensions, EOC-Bench structures questions into three temporally grounded categories: Past, Present, and Future.

- **Past:** The Past category evaluates MLLMs' ability to perceive and understand the historical dynamics of objects, a skill crucial for enhancing long-term task execution. As illustrated in Fig. 1, this capability is further subdivided into four types: Object State Retrospection, Object Location Retrospection, Object Relationship Evolution, and Absolute Time Perception. The last is particularly vital, as accurate timestamp awareness of model can contextualize interactions and temporal changes, which has received little attention in previous benchmarks.

- **Present:** The Present category test MLLMs' perception of scene information at the current moment. Importantly, resolving these questions often require more than just observing the current frame; a comprehensive understanding of the entire video is necessary for accuracy. As shown in Fig. 1, in addition to common abilities such as Immediate State Recognition, Purpose and Function Inference, and Object Relationship, we introduce Anomaly Perception to handle embodied tasks in specific scenarios. This capability tests whether MLLMs can avoid being misled by counterintuitive arrangements within the scene and answer questions based on factual information about the objects.

- **Future:** By observing the world, human can not only understand past events but also predict future occurrences. The capability to foresee future events in objects is crucial for avoiding dangers and adapting to new situations. For instance, as shown in the State Change Prediction part in Fig. 1, if a model identifies a heat-sensitive object near a heat source, it can anticipate temperature changes and alert people to move the object to prevent hazards. Future prediction types are divided into State Change Prediction, Dynamic Relationship Prediction, and Trajectory and Motion Prediction, assessing MLLMs' proficiency in forecasting dynamic interactions and movements.

To ensure a comprehensive evaluation, we develop a mixed-format annotation framework featuring diverse question types (*e.g.*, true/false, single-choice, multiple-choice and open-ended questions), as visualized in Fig. 1. Specially, for open-ended questions, we focus on continuous temporal analysis and introduce a multi-scale temporal accuracy metric to quantitatively assess temporal perception performance. Additionally, traditional text-based prompts for object referencing often fail

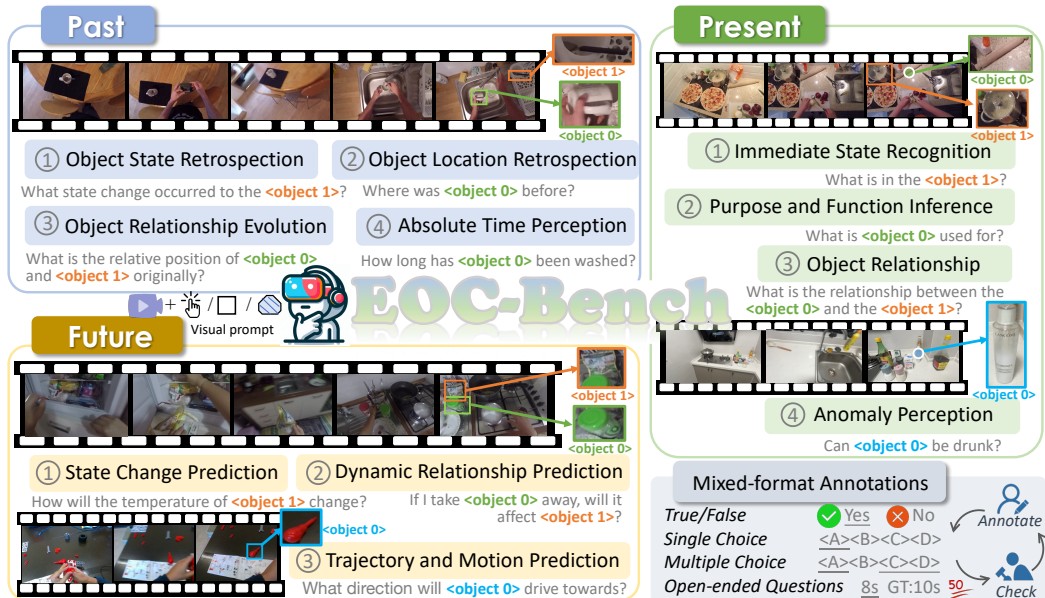

Figure 1: **Overview of** `EOC-Bench`. `EOC-Bench` assesses Embodied Object Cognition capabilities of MLLMs in egocentric videos across three dimensions - Past, Present, and Future - encompassing **11** categories. `EOC-Bench` includes **3,277** object-level QA pairs utilizing a mixed-format ***human-in-the-loop*** annotation framework across diverse tasks and conditions. `EOC-Bench` aims to reveal the limitations of MLLMs and promote the development of robust egocentric cognition systems.

to clearly specify objects in dynamic egocentric scenes. Descriptions like "the leftmost bowl" become meaningless when containers are rearranged during washing, and "the spoon" lacks clarity among multiple candidates in the kitchen. To address this issue, we introduce visual referencing prompts, including point, box and mask, as shown in Fig. 1, which provide persistent, unequivocal object references while preserving the spatiotemporal context essential for precise object comprehension. The final benchmark includes 3,277 question-answer pairs, covering 11 fine-grained evaluation dimensions and 3 object referencing types. We have conducted a meticulous *human-in-the-loop* labeling process, followed by comprehensive cross-checking and verification to ensure quality.

Building upon our `EOC-Bench` benchmark, we systematically evaluate the egocentric object cognition capabilities of a range of MLLMs, including both open-source and proprietary general-purpose models [3, 4, 1, 20, 21], as well as specialized object-level MLLMs [12, 22, 23]. Notably, all mainstream MLLMs exhibit clear deficiencies in object-level temporal perception, particularly concerning absolute temporal awareness, where they significantly lags behind human-level performance, emphasizing its difficulty and relevance for our community.

## 2 Related Work

### 2.1 General Video Understanding Benchmarks

With the advancement of MLLMs [3, 24, 1, 21, 25, 26, 27, 4, 2, 28, 29, 30, 31, 32, 33, 34], which have demonstrated strong visual understanding and reasoning capabilities, there is an increasing emphasis on comprehensively and systematically evaluating their video understanding abilities [11, 35, 36, 37]. Existing video understanding benchmarks primarily focus on general-purpose video comprehension tasks, such as action recognition [38, 39, 40], video caption [41, 42, 43], temporal grounding [38, 44, 13], temporal reasoning [45, 46, 47, 44], long video understanding [48, 49, 11, 50, 36, 51], video referring [12, 52] and expert-level reasoning [53, 54]. For instance, Video-MME [35] conducts an extensive evaluation of MLLMs across a variety of video-related tasks, such as recognition and perception. Similarly, MVBench [11] introduces an innovative framework for constructing spatial-temporal tasks. However, general VideoQA benchmarks predominantly focus on YouTube videos that capture everyday life, human actions, and movies, often neglecting to include egocentric videos and embodied-specific QA formats.

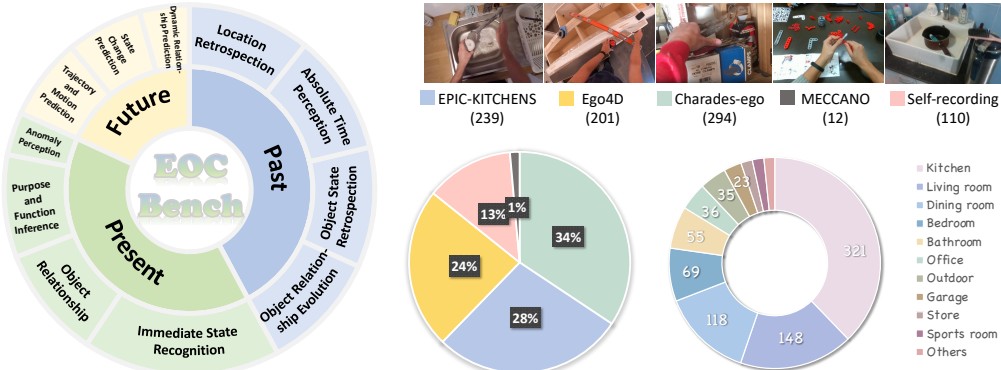

(a) Overview of EOC-Bench dimensions    (b) Video source distribution    (c) Number of various scenario categories

Figure 2: **Overall data distribution of** `EOC-Bench`. (a) `EOC-Bench` encompasses three temporal dimensions: Past, Present, and Future, comprehensively evaluating 11 embodied cognitive abilities. (b) The dataset comprises videos from four distinct open video sources as well as self-recorded videos. (c) It spans a wide range of scenarios, offering a rich diversity of contexts for analysis.

## 2.2 Embodied Video Understanding Benchmarks

In embodied scenarios, VideoQA-based evaluations serve as effective tools for assessing a model's comprehension of its environment and tasks. Datasets such as ScanQA [14], SQA3D [15], and Env-QA [55] are typically used for traditional scene question answering, characterized by a closed vocabulary. These datasets often exhibit a strong text bias and offer a relatively limited variety of question forms. On the other hand, RoboVQA [16], EgoPlan-Bench [56] & EgoPlan-Bench2 [57], and PCA-Bench [58] are introduced test the task-planning abilities of MLLMs. EgoSchema [59] utilizes first-person footage from Ego4D [60] to enable video reasoning tasks. More recently, VSI-Bench [18] has been developed to specifically evaluate visual-spatial intelligence in MLLMs, and STI-Bench [61] has been further developed to evaluate the spatial-temporal world understanding. OpenEQA [17] and ECBench [19] systematically investigate the embodied indoor cognition of MLLMs, providing a wider scope of evaluation diversity. However, these benchmarks mainly focus on static scene exploration, neglecting dynamic first-person operational interactions involving hand and object movements. Furthermore, while these benchmarks try to assess models' embodied cognitive abilities through text-based object referencing, they fall short of adequately evaluating models' capabilities in object-level spatiotemporal reasoning, which is crucial for real-world interactions. In contrast, we have meticulously crafted `EOC-Bench` to systematically analyze the object-level embodied cognition of MLLMs in complex dynamic operational scenes.

## 3 EOC-Bench

### 3.1 Overview

As illustrated in Fig. 2, we introduce `EOC-Bench`, a meticulously crafted benchmark designed to quantitatively assess the object cognition abilities of MLLMs using dynamic egocentric videos. `EOC-Bench` comprises 3,277 question-answer pairs derived from 656 real-world videos. These videos are sourced from four publicly available first-person datasets: EPIC-KITCHENS [62], Ego4D [60], Charades-ego [63], and MECCANO [64], as well as our self-recorded videos captured in various environments. `EOC-Bench` includes three dimensions: Past, Present and Future, with a total of 11 tasks aimed at evaluating a model's object comprehension capabilities including memory, perception and knowledge in ego-centric world. Notably, to achieve accurate object referencing in dynamic scenarios, we introduce three types of visual object prompts: bounding boxes, points and masks.

### 3.2 Benchmark Construction

#### 3.2.1 Video Collection

Our benchmark integrates four established egocentric video datasets: EPIC-KITCHENS [62], which features kitchen-related scenarios; Ego4D [60], encompassing a broad array of daily activities;

Charades-ego [63], capturing activity instances across various rooms; and MECCANO [64], depicting industrial-like environments where participants construct toy models. These datasets collectively cover both indoor and outdoor environments, covering a wide spectrum of activities. To enhance scenario diversity, we develop a stratified sampling strategy. Initially, we sample 1,000 videos each from Charades-ego [63] and Ego4D [60] and annotate them for scene categories using Qwen2-VL-72B [24]. We further enhance scene diversity by randomly sampling from videos featuring the same setting, followed by thorough manual quality control to eliminate clips with low information. This process results in 294 high-quality videos from Charades-ego and 201 from Ego4D. For datasets like EPIC-KITCHENS and MECCANO with uniform scenes, we randomly choose 239 and 12 representative videos, respectively. All selected videos are uniformly trimmed to durations of 3-10 minutes for efficient annotation. To address gaps in existing datasets, we self-curate 110 videos capturing three under-represented domains: anomaly perception, physical world dynamics, and electrical appliance operation. *To ensure diversity, 5 volunteers contribute to the collection process.*

### 3.2.2 Capability Taxonomy

Drawing inspiration from established general VideoQA benchmarks [11, 36], we propose a hierarchical taxonomy to systematically characterize embodied object cognition capabilities, as shown in Figure 2-(a). `EOC-Bench` comprehensively encompasses three temporal dimensions of first-person video understanding: Past, Present, and Future.

**Past.** This dimension assesses a model's ability to perceive and interpret the temporal dynamics of objects, a critical skill for long-term and complex operations. This capability enables models to enhance their current understanding by integrating insights from past interactions. The Past dimension is specifically divided into four categories:

- **Object State Retrospection (OSR):** Evaluates the capability to monitor changes in object attributes including color, shape, size, posture, temperature, and motion.
- **Object Location Retrospection (OLR):** Measures historical positioning accuracy across multiple granularity: macro-level (room-scale), meso-level (platform/container positioning), and micro-level (precise location).
- **Object Relationship Evolution (ORE):** Examines changes in object relationships, encompassing spatial relationships, motion state dynamics, and temporal sequence relationships.
- **Absolute Time Perception (ATP):** Assesses absolute time cognition precision through two key aspects, including pinpointing specific time points and understanding time durations.

**Present.** This category focuses on evaluating MLLMs' ability to understand current scenes, with a focus on the perceptual abilities. Crucially, while emphasizing immediate perception of object states and environmental conditions, some questions necessitate integration of information from preceding frames, demanding a comprehensive understanding of the video for accurate responses. This aspect is categorized into four types:

- **Immediate State Recognition (ISR):** Evaluates the model's ability to identify the current state of objects, including attributes such as material, shape, functional state, surface condition, pose, motion state, and temperature.
- **Object Relationship (OR):** Analyzes inter-object dynamics, including spatial, functional, or comparative relationships between existing objects.
- **Purpose and Function Inference (PFI):** Requires deducing the potential uses or functions of objects based on their external characteristics, materials, configurations, and the contextual scenarios in which they are observed.
- **Anomaly Perception (AP):** Measures the model's proficiency in detecting unusual or incongruous visual inputs, with an emphasis on counter-sense co-occurrence. For instance, Fig. 1 illustrates a scenario where a cosmetic product is placed in an atypical setting, such as a kitchen, to assess common sense interference in visual interpretation.

**Future.** In embodied intelligence systems, predictive capabilities extend beyond mere observation, empowering proactive adaptation to environmental changes. The capability to foresee future events is crucial for avoiding hazards and flexibly adapting to changing circumstances. This dimension

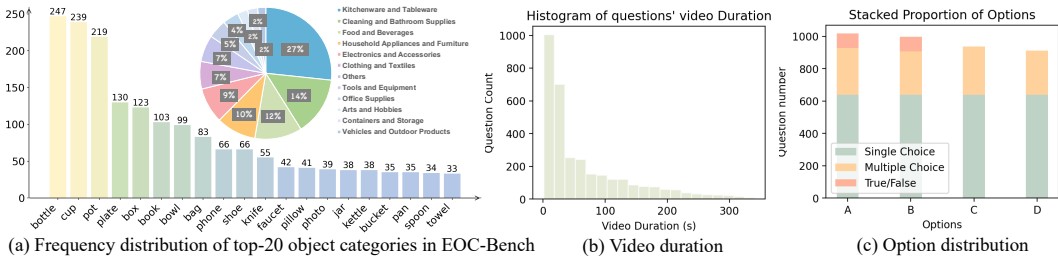

(a) Frequency distribution of top-20 object categories in EOC-Bench

(b) Video duration

(c) Option distribution

Figure 3: **Statistic analysis** of `EOC-Bench`: (a) substantial diversity in object categories and usage taxonomies, (b) a wide range of video durations correlated with question count, and (c) a balanced distribution of response options across each question type.

relies on the model's ability to utilize physical laws and common sense knowledge for prediction and inference. This dimension is divided into three categories:

- **Trajectory and Motion Prediction (TMP):** Anticipates the future path or dynamic motion changes of an object based on its current motion and location, enabling models to understand and interact with moving objects more effectively.
- **State Change Prediction (SCP):** Predicts future changes in an object's state due to ongoing actions or environmental fluctuations, enabling preemptive response to imminent changes.
- **Dynamic Relationship Prediction (DRP):** Foresees potential alterations in inter-object relationships, aiding in the prevention of upcoming collisions or other interactions.

### 3.2.3 Construction of Question-Answer Pairs

To ensure the high quality of our benchmark, we have developed a sophisticated *human-in-the-loop* data curation pipeline specifically for the creation of `EOC-Bench`, and we recruit 10 highly trained university students as annotators to participate in the annotation process. Our methodology adopts a category-independent approach, assigning volunteers a predetermined number of tasks related to various cognitive abilities. This strategy guarantees a balanced representation of question-answer (QA) pairs, covering both rare and common cognitive abilities. `EOC-Bench` features a mixed-format annotation framework with four types of labeling: True/False, Single-Choice, Multiple-Choice questions, which require explicit options, while Open-Ended questions are crafted to primarily focus on absolute timestamps information for temporal perception abilities.

Despite leveraging human-annotated data sources and implementing a meticulously designed QA generation protocol, certain ambiguities and errors may still occur, such as visual prompt offsets, omissions, and ambiguous options. To address these issues, *a thorough filtering process is carried out post-labeling*. This involves rigorous cross-checking and verification among annotators to ensure both format accuracy and content validity.

### 3.2.4 Evaluation Metrics

Our `EOC-Bench` includes diverse question types: True/False ($\mathcal{TF}$), Single-Choice Answer ($\mathcal{SCA}$), Multiple-Choice Answer ($\mathcal{MCA}$) and Open-Ended Questions ($\mathcal{OQ}$). Following established practices [11, 35], we adopt conventional *Accuracy* based on exact matches for the first three tasks. For Open-Ended Questions, which require assessing open-ended continuous temporal predictions, we introduce a novel metric, *Multi-Scale Temporal Accuracy* ($\mathcal{MSTA}$), to accurately evaluate $\mathcal{OQ}$ tasks.

Specially, we develop a relative error percentage tolerance mechanism to accommodate varying error tolerance across different time durations, whether long or short periods. Given a ground truth timestamp $T_{gt}$ and a predicted time $T_{pred}$, we first calculate the absolute deviation $\Delta T = |T_{pred} - T_{gt}|$. We then establish dynamic error margins using relative percentage thresholds $C = \{1\%, 10\%, 20\%, 30\%\}$, setting scale-adaptive boundaries $\{\alpha \cdot T_{gt} \mid \alpha \in C\}$. These thresholds are derived from human error analysis, which is detailed in the Appendix. A prediction satisfies threshold $\alpha$ when $\Delta T \leq \alpha \cdot T_{gt}$. The final $\mathcal{MSTA}$ score is computed by averaging performance across temporal scales using:

$$\mathcal{MSTA} = \frac{1}{4} \sum_{\alpha \in C} \mathbb{1}\left(\Delta T \leq \alpha \cdot T_{gt}\right). \tag{1}$$

By utilizing various thresholds, $\mathcal{MSTA}$ strikes a balance between strictness and flexibility: lower thresholds demand precise alignment, while higher thresholds allow for variability in responses.

### 3.3 Benchmark Statistics

`EOC-Bench` comprises **3,277** QA pairs, systematically evaluating MLLMs across **11** cognitive perspectives. These include **1,422** questions focused on the Past dimension, **1,348** on the Present, and **507** on the Future. Each question is associated with one or more objects, and the corresponding visual prompts are annotated on the final frame of the video. The benchmark incorporates a wide array of object types, encompassing **728** categories that cover various usage scenarios. The category distribution, along with the top 20 categories, is displayed in Fig. 3-(a). Additionally, Fig. 3-(b) illustrates the distribution of average video durations, which vary widely from several seconds to over six minutes. To maintain an even probability distribution for each response option, we rearranged the order of different answer types, as depicted in Fig. 3-(c).

## 4 Experiment

### 4.1 Experimental Setup

Based on `EOC-Bench`, we comprehensively evaluate a diverse range of general-purpose MLLMs, including both proprietary MLLMs and open-source models. For *proprietary MLLMs*, we evaluate GPT-4o [3], GPT-4o-mini [3] and Gemini-2.0-flash [4]. Among *open-source MLLMs*, we test Qwen2.5-VL [1], InternVL2.5 [20], VideoLLaMA2&3 [21, 25], LLaVA-OneVision [27], LLaVA-Video [65], NVILA [66], LongVA [31] and VideoLLaVA [28]. Additionally, we assess the object-focused MLLMs including VideoRefer [12], ViP-LLaVA [23], Osprey [22] and SPHINX-V [67]. For all models, we perform zero-shot inference to assess their object cognition capabilities using their default settings. More detailed configurations are provided in the Appendix.

### 4.2 Main Results

In this section, we provide a detailed performance comparison and analysis. Table 2 reports the main experimental results.

**Baselines.** The "Random" entry in the first row denotes random guessing. For multiple-choice answers, we randomly select the number of options and the corresponding choices. For open-ended questions in Absolute Time Perception (ATP) task within the Past dimension, values are randomly selected between 0 and video length. Additionally, we also assess human performance on `EOC-Bench` using video input with three volunteers.

**Proprietary MLLMs.** Despite a significant performance gap compared to human capabilities, the leading proprietary model, GPT-4o [3], delivers commendable results with 61.83%. GPT-4o [3] successfully meets the passing criteria across various subtasks, showcasing its potential in multiple domains. However, the model faces challenges in the Past dimension, particularly with Absolute Time Perception (ATP) and Object Relationship Evolution (ORE), even when timestamps are provided for each frame. This indicates the model's limited capacity to perceive and remember temporal changes. The difficulties encountered by GPT-4o [3] in these areas underscore a significant opportunity for improvement, highlighting the need for advancements in temporal awareness and memory retention.

**Open-source MLLMs.** Top-tier open-source models, like InternVL2.5-78B [20], still show a noticeable gap compared to closed-source models, trailing GPT-4o [3] by 9.5%. Other state-of-the-art Video-LLMs on existing benchmarks, such as Qwen2.5-VL [1], VideoLLaMA3 [21], and NVILA [66], underperform on our tasks, particularly in Object Relationship Evolution (ORE) and Absolute Time Perception (ATP). A substantial number of these models are tagged with grey marks, indicating significant limitations in their memory recall capabilities.

**Object-level MLLMs.** Object-level MLLMs, such as the recent VideoRefer [12], outperform many competitive models, highlighting the effectiveness of the object-level representation learning. However, they still face challenges in the Object Relationship Evolution (ORE) task when dealing with dense, similar objects in complex operational scenes, and in the Absolute Time Perception (ATP) task with dynamic temporal changes. Given the scarcity of open-source object-level video MLLMs, we also evaluated some image-level MLLMs, like ViP-LLaVA [23], Osprey [22] and SPHINX-V [67].

| Method | Input | Mean | Past | | | | | Present | | | | | Future | | | |
|---|---|---|---|---|---|---|---|---|---|---|---|---|---|---|---|---|
| | | | OSR | OLR | ORE | ATP | Mean | ISR | OR | PFI | AP | Mean | TMP | SCP | DRP | Mean |
| Random | - | 24.87 | 29.36 | 26.56 | 26.46 | 10.92 | 24.75 | 26.30 | 23.30 | 21.29 | 26.47 | 24.41 | 27.80 | 21.96 | 34.09 | 26.43 |
| Human | - | 94.63 | 96.95 | 93.49 | 94.71 | 74.30 | 90.67 | 99.33 | 98.23 | 96.77 | 93.14 | 97.99 | 95.12 | 95.79 | 90.91 | 94.67 |
| *Proprietary Multimodal Foundation Models* | | | | | | | | | | | | | | | | |
| Gemini-2.0-flash [4] | 32f | 45.45 | 50.42 | 34.42 | 22.84 | 10.32 | 29.78 | 61.98 | 56.36 | 69.35 | 51.96 | 61.50 | 52.20 | 54.70 | 48.86 | 52.53 |
| GPT-4o-mini* [3] | 32f | 49.47 | 53.26 | 52.35 | 29.68 | 21.10 | 39.47 | 58.46 | 49.26 | 67.74 | 58.82 | 58.31 | 56.59 | 50.00 | 54.55 | 53.45 |
| Gemini-2.0-flash* [4] | 32f | 57.38 | 63.46 | 65.10 | 32.56 | 28.60 | 47.87 | 68.84 | 57.52 | 69.68 | 65.69 | 65.95 | 58.54 | 64.02 | 57.95 | 60.75 |
| GPT-4o* [3] | 32f | 61.83 | 66.04 | 71.93 | 46.56 | 34.46 | 54.91 | 71.46 | 52.85 | 78.18 | 62.75 | 67.32 | 69.61 | 68.69 | 68.97 | 69.11 |
| *Open-Source Multimodal Foundation Models* | | | | | | | | | | | | | | | | |
| VideoLLaVA-7B [28] | 8f | 34.11 | 31.86 | 37.94 | 27.58 | 13.14 | 27.97 | 41.04 | 35.10 | 40.97 | 37.25 | 39.24 | 40.98 | 31.78 | 44.32 | 37.67 |
| LongVA-7B [31] | 32f | 35.34 | 36.84 | 43.36 | 17.83 | 15.32 | 28.69 | 38.19 | 36.58 | 48.06 | 42.16 | 40.36 | 39.02 | 42.06 | 40.91 | 40.63 |
| NVILA-8B [66] | 32f | 37.69 | 37.40 | 46.61 | 20.89 | 12.09 | 29.69 | 44.39 | 41.59 | 49.03 | 46.08 | 44.88 | 42.44 | 38.32 | 44.32 | 41.03 |
| VideoLLaMA2.1-7B [25] | 16f | 37.74 | 44.88 | 42.82 | 19.22 | 11.64 | 30.08 | 47.24 | 37.17 | 51.94 | 39.22 | 45.18 | 40.00 | 36.92 | 44.32 | 39.45 |
| Qwen2.5-VL-3B [1] | 1fps | 38.17 | 38.78 | 48.78 | 23.96 | 7.66 | 30.34 | 49.92 | 38.94 | 45.16 | 38.24 | 45.18 | 42.93 | 36.57 | 50.00 | 41.45 |
| VideoLLaMA3-2B [21] | 1fps | 38.41 | 37.12 | 46.88 | 21.17 | 11.26 | 29.57 | 49.92 | 43.36 | 48.39 | 38.24 | 47.03 | 43.41 | 36.11 | 43.18 | 40.28 |
| LLaVA-OV-7B [27] | 32f | 40.46 | 40.72 | 45.53 | 22.84 | 9.53 | 30.15 | 54.10 | 43.07 | 52.58 | 46.08 | 50.37 | 47.32 | 37.38 | 46.59 | 43.00 |
| VideoLLaMA2-72B [25] | 16f | 41.55 | 43.77 | 51.22 | 24.23 | 6.46 | 32.03 | 50.08 | 37.46 | 58.06 | 45.10 | 48.37 | 49.27 | 50.47 | 51.14 | 50.10 |
| LLaVA-Video-7B [65] | 32f | 41.82 | 44.32 | 48.51 | 22.56 | 9.76 | 31.82 | 54.27 | 43.66 | 55.81 | 49.02 | 51.56 | 45.85 | 40.65 | 47.73 | 43.98 |
| Qwen2.5-VL-7B [1] | 1fps | 43.13 | 47.37 | 46.34 | 21.45 | 8.18 | 31.38 | 57.29 | 44.54 | 59.35 | 49.02 | 53.93 | 48.78 | 46.30 | 46.59 | 47.35 |
| InternVL2.5-8B [20] | 32f | 45.15 | 45.71 | 54.47 | 39.00 | 9.76 | 37.87 | 55.44 | 48.97 | 54.84 | 41.18 | 52.60 | 49.76 | 38.79 | 53.41 | 45.76 |
| VideoLLaMA3-7B [21] | 1fps | 46.04 | 45.15 | 52.85 | 24.51 | 15.54 | 35.00 | 57.96 | 48.67 | 62.58 | 49.02 | 56.01 | 52.20 | 49.54 | 48.86 | 50.49 |
| LLaVA-OV-72B [27] | 32f | 47.88 | 46.81 | 50.95 | 26.46 | 12.91 | 34.81 | 64.15 | 51.33 | 64.52 | 49.02 | 59.87 | 58.05 | 46.73 | 54.55 | 52.66 |
| LLaVA-Video-72B [65] | 32f | 49.59 | 49.03 | 56.91 | 26.74 | 24.02 | 39.59 | 63.32 | 47.20 | 63.87 | 50.00 | 58.38 | 56.10 | 55.14 | 47.73 | 54.24 |
| Qwen2.5-VL-72B [1] | 1fps | 49.87 | 51.25 | 51.22 | 40.11 | 8.48 | 38.41 | 61.31 | 47.79 | 67.10 | 57.84 | 58.98 | 56.10 | 60.65 | 54.55 | 57.76 |
| InternVL2.5-38B [20] | 32f | 52.31 | 55.40 | 59.62 | 30.92 | 10.89 | 39.89 | 64.15 | 54.28 | 71.29 | 64.71 | 63.35 | 60.98 | 54.67 | 57.95 | 57.79 |
| InternVL2.5-78B [20] | 32f | 52.33 | 53.46 | 63.96 | 33.15 | 12.01 | 41.35 | 66.67 | 50.74 | 67.10 | 52.94 | 61.72 | 67.80 | 50.47 | 54.55 | 58.19 |
| *Object-level Multimodal Models* | | | | | | | | | | | | | | | | |
| Osprey-7B [22] | 1f | 27.36 | 22.71 | 20.33 | 15.88 | 7.41 | 16.78 | 42.88 | 29.50 | 32.58 | 29.41 | 36.13 | 39.51 | 30.37 | 28.41 | 33.73 |
| SPHINX-V-13B [67] | 1f | 29.21 | 25.48 | 23.31 | 13.37 | 3.83 | 16.79 | 41.71 | 31.27 | 44.19 | 39.22 | 39.47 | 41.46 | 31.02 | 39.77 | 36.74 |
| ViP-LLaVA-7B [23] | 1f | 32.82 | 35.73 | 36.86 | 17.55 | 8.26 | 25.00 | 42.88 | 35.99 | 46.45 | 26.47 | 40.73 | 34.63 | 29.91 | 40.91 | 33.73 |
| VideoRefer-7B [12] | 16f | 40.44 | 47.37 | 55.01 | 23.40 | 10.59 | 34.69 | 48.91 | 39.82 | 53.55 | 38.24 | 46.88 | 41.95 | 35.51 | 43.18 | 39.45 |

Table 2: **Performance of representative MLLMs on** `EOC-Bench`. The best results are marked with orange . The results below random guess are marked with grey . Entries in grey indicate image-level methods that use only the last frame as input. ∗: We manually added a timestamp before each frame. [Nf] denotes that the model takes N frames uniformly sampled from a video as input.

While these models underperform in the Past dimension, which requires memory of previous frames, they still deliver reasonably performance in the Present and Future dimensions.

## 4.3 Analysis Across Different Question Types

We conduct an analysis of the models' results across different question types to facilitate a more comprehensive horizontal and vertical examination, as illustrated in Table 3.

**Smaller MLLMs Often Struggle with Multiple-Choice Questions.** Many MLLMs face challenges in answering multiple-choice questions ($\mathcal{MCA}$), often scoring lower than random guess (indicated by a grey mark). This issue is particularly evident in smaller models, those with 7B parameters or fewer. We surmise that these smaller models have overfitted to simple single-choice questions during training, hindering their ability to follow instructions for handling questions with multiple options.

**Few MLLMs are Time-sensitive.** The $\mathcal{OQ}$ metric, which measures the model's ability to perceive past time, indicates that the some models perform below random guessing levels, with 9/21. Even the strongest open-source model scores only 24.02%, just 13.1% above random chance. This underscores a crucial capability that is lacking in most models, yet is essential in the field of embodied AI.

**Larger MLLMs Excel in Handling Future-Oriented Problems.** Future-oriented tasks demand a combination of commonsense reasoning and extensive knowledge. Our observations indicate that as the size of the model increases, so does its reasoning capability. For instance, Qwen2.5-VL [1] with 3B, 7B, and 72B parameters, as well as VideoLLaMA3 [21] with 2B and 7B parameters, demonstrate significantly improved performance in these tasks. This trend suggests that larger MLLMs are better equipped to tackle problems that require forward-thinking and predictive reasoning, due to their enhanced capacity to integrate and process complex patterns of information.

**Past-Oriented Questions Pose Greater Challenges to MLLMs.** Through a comparative analysis of similar problem types, we discover that models generally perform worse on questions related to past events compared to other categories. While smaller models may grapple with future-oriented problems, larger models often fall short when addressing past-oriented questions. This difficulty in accurately recalling and processing past information is a prevalent issue among current MLLMs, indicating a significant area for improvement in their design and training.

| Method | Input | Mean | | | | Past | | | | Present | | | Future | | |
|---|---|---|---|---|---|---|---|---|---|---|---|---|---|---|---|
| | | $\mathcal{SCA}$ | $\mathcal{MCA}$ | $\mathcal{TF}$ | $\mathcal{OQ}$ | $\mathcal{SCA}$ | $\mathcal{MCA}$ | $\mathcal{TF}$ | $\mathcal{OQ}$ | $\mathcal{SCA}$ | $\mathcal{MCA}$ | $\mathcal{TF}$ | $\mathcal{SCA}$ | $\mathcal{MCA}$ | $\mathcal{TF}$ |
| Random | - | 26.27 | 18.34 | 50.00 | 10.92 | 29.51 | 20.56 | 50.00 | 10.92 | 24.72 | 14.95 | 50.00 | 23.53 | 18.60 | 50.00 |
| *Proprietary Multimodal Foundation Models* | | | | | | | | | | | | | | | |
| GPT-4o* [3] | 32f | 69.03 | 54.44 | 63.86 | 34.46 | 67.00 | 49.06 | 55.56 | 34.46 | 69.29 | 58.08 | 68.85 | 72.76 | 63.95 | 61.46 |
| GPT-4o-mini* [3] | 32f | 57.31 | 32.80 | 58.76 | 21.10 | 52.07 | 26.26 | 44.44 | 21.10 | 60.79 | 41.24 | 67.14 | 58.20 | 34.88 | 54.08 |
| Gemini-2.0-flash* [4] | 32f | 68.68 | 28.49 | 63.28 | 28.60 | 66.15 | 20.14 | 44.44 | 28.60 | 70.66 | 37.63 | 71.43 | 68.11 | 34.88 | 59.18 |
| *Open-Source Multimodal Foundation Models* | | | | | | | | | | | | | | | |
| VideoLLaVA-7B [28] | 8f | 41.55 | 11.11 | 54.80 | 13.14 | 41.24 | 8.36 | 33.33 | 13.14 | 42.34 | 14.95 | 58.57 | 39.63 | 11.63 | 54.08 |
| VideoLLaMA2.1-7B [25] | 16f | 47.67 | 9.01 | 52.78 | 11.64 | 45.40 | 8.36 | 55.56 | 11.64 | 51.29 | 9.28 | 50.00 | 41.27 | 10.81 | 54.46 |
| VideoLLaMA2-72B [25] | 16f | 53.15 | 13.15 | 51.67 | 6.46 | 52.21 | 5.23 | 55.56 | 6.46 | 52.77 | 22.68 | 51.43 | 56.63 | 18.92 | 51.49 |
| LongVA-7B [31] | 32f | 41.73 | 16.40 | 54.24 | 15.32 | 41.24 | 9.06 | 44.44 | 15.32 | 41.97 | 23.71 | 61.43 | 42.11 | 24.42 | 50.00 |
| NVILA-8B [66] | 32f | 49.73 | 0.53 | 55.37 | 12.09 | 47.41 | 0 | 66.67 | 12.09 | 51.85 | 1.55 | 57.14 | 48.30 | 0 | 53.06 |
| InternVL2.5-8B [20] | 32f | 54.95 | 23.99 | 57.63 | 9.76 | 55.11 | 22.30 | 55.56 | 9.76 | 56.55 | 26.80 | 62.86 | 49.23 | 23.26 | 54.08 |
| InternVL2.5-38B [20] | 32f | 64.45 | 26.28 | 62.71 | 10.89 | 62.67 | 10.10 | 55.56 | 10.89 | 66.70 | 43.30 | 67.14 | 61.30 | 41.86 | 60.20 |
| InternVL2.5-78B [20] | 32f | 64.32 | 26.63 | 61.58 | 12.01 | 62.55 | 16.38 | 55.56 | 12.01 | 65.13 | 40.21 | 68.57 | 65.94 | 30.23 | 57.14 |
| LLaVA-OV-7B [27] | 32f | 54.18 | 0 | 57.63 | 9.53 | 49.43 | 0 | 55.56 | 9.53 | 58.67 | 0 | 61.43 | 50.77 | 0 | 55.10 |
| LLaVA-OV-72B [27] | 32f | 61.32 | 12.17 | 61.02 | 17.22 | 55.49 | 1.74 | 77.78 | 17.22 | 66.05 | 22.68 | 67.14 | 59.75 | 23.26 | 55.10 |
| LLaVA-Video-7B [65] | 32f | 56.23 | 0 | 57.06 | 9.76 | 52.33 | 0 | 55.56 | 9.76 | 59.78 | 0 | 67.14 | 53.87 | 0 | 50.00 |
| LLaVA-Video-72B [65] | 32f | 62.73 | 10.23 | 60.45 | 24.02 | 59.27 | 2.44 | 66.67 | 24.02 | 65.22 | 18.56 | 62.86 | 62.85 | 17.44 | 58.16 |
| Qwen2.5-VL-3B [1] | 1fps | 50.50 | 1.44 | 56.67 | 7.66 | 50.44 | 0 | 66.67 | 7.66 | 51.85 | 3.09 | 58.57 | 46.25 | 2.67 | 54.46 |
| Qwen2.5-VL-7B [1] | 1fps | 54.25 | 13.67 | 62.22 | 8.18 | 49.81 | 6.27 | 66.67 | 8.18 | 58.39 | 23.71 | 68.57 | 51.35 | 16.00 | 57.43 |
| Qwen2.5-VL-72B [1] | 1fps | 61.45 | 27.16 | 54.44 | 8.48 | 57.12 | 21.60 | 33.33 | 8.48 | 63.19 | 34.54 | 61.43 | 66.07 | 29.33 | 51.49 |
| VideoLLaMA3-2B [21] | 1fps | 49.98 | 3.70 | 57.06 | 11.26 | 47.29 | 1.05 | 55.56 | 11.26 | 60.89 | 6.19 | 64.29 | 45.68 | 6.90 | 52.04 |
| VideoLLaMA3-7B [21] | 1fps | 57.15 | 17.27 | 55.00 | 15.54 | 52.33 | 9.41 | 44.44 | 15.54 | 60.89 | 26.29 | 62.86 | 56.46 | 24.00 | 50.50 |
| VideoRefer-7B [12] | 16f | 54.27 | 0 | 54.24 | 10.59 | 57.12 | 0 | 55.56 | 10.59 | 54.43 | 0 | 60.00 | 46.75 | 0 | 50.00 |

Table 3: **Performance of representative MLLMs across different question types:** $\mathcal{SCA}$ (Single-Choice Answer), $\mathcal{MCA}$ (Multi-Choice Anwer), $\mathcal{TF}$ (True/False), $\mathcal{OQ}$ (Open-Ended Question). The best results are marked with orange. The results below random guess are marked with grey. *: We manually added a timestamp before each frame.

| # Frames | Mean | | | | Past | | | | Present | | | | Future | | | |
|---|---|---|---|---|---|---|---|---|---|---|---|---|---|---|---|---|
| | 1f | 8f | 32f | $\gamma\uparrow$ | 1f | 8f | 32f | $\gamma\uparrow$ | 1f | 8f | 32f | $\gamma\uparrow$ | 1f | 8f | 32f | $\gamma\uparrow$ |
| GPT-4o* [3] | 49.6 | 58.6 | 61.8 | 24.6 | 36.8 | 50.6 | 54.9 | 49.2 | 60.2 | 64.7 | 67.3 | 11.8 | 58.0 | 65.2 | 69.1 | 19.1 |
| Gemini-2.0-flash* [4] | 47.8 | 51.2 | 57.4 | 20.1 | 29.9 | 37.7 | 47.9 | 60.2 | 64.7 | 63.4 | 66.0 | 2.0 | 53.1 | 57.0 | 60.8 | 14.5 |
| InternVL2.5-78B [20] | 47.6 | 51.3 | 52.3 | 9.9 | 33.1 | 38.9 | 41.4 | 24.0 | 59.8 | 64.3 | 61.7 | 3.2 | 55.8 | 56.4 | 58.2 | 4.3 |
| VideoLLaMA3-7B [21] | 42.1 | 45.5 | 46.2 | 10.4 | 28.5 | 34.3 | 36.1 | 26.7 | 54.6 | 55.0 | 55.1 | 0.9 | 46.5 | 49.7 | 50.5 | 8.6 |

Table 4: **Performance of representative MLLMs with varying input frames.** '1f' denotes using only the last frame, while '8f/32f' refers to frames that are uniformly sampled, including the last frame. $\gamma$ represents the rate of increase in performance from '1f' to '32f'.

## 4.4 Multi-Frame Gain

We assess the multi-frame gain for frames 1, 8, and 32 within EOC-Bench. The strong proprietary MLLMs, GPT-4o [3] and Gemini-2.0-flash [4], exhibits a substantial performance boost, gaining 24.6% and 20.1% when moving from single-frame input to 32-frame input setting. This improvement is particularly pronounced in past-oriented tasks, with an improvement of 49.2% and 60.2%. These findings underscore the critical role of multi-frame reasoning in the EOC-Bench, especially for memory recall tasks. The ability to access information from previous frames can significantly enhance both current and future understanding. Other open-source models, such as InternVL2.5-78B [20], and VideoLLaMA3-7B [21], demonstrate similar trends. However, their ability to effectively process multiple frames is comparatively weaker, resulting in less pronounced performance improvements. This highlights the potential benefits of enhancing multi-frame processing capabilities in MLLMs to achieve more substantial performance gains across a variety of tasks.

## 5 Conclusion

In this paper, we presented EOC-Bench, an innovative benchmark aimed at evaluating the embodied, object-level cognition capabilities of MLLMs. EOC-Bench thoroughly assesses MLLMs within the scenes involving dynamic egocentric interactions across three temporal dimensions: Past, Present and Future. To ensure high quality, we developed a mixed-format human-in-the-loop annotation framework and introduced a multi-scale temporal accuracy metric to enhance the precision of open-ended questions. Extensive evaluations conducted on EOC-Bench across a range of proprietary and open-source models, have revealed that many MLLMs face challenges in effectively performing embodied object cognition tasks, particularly in recalling and processing past information as well as in absolute time perception. We hope EOC-Bench will drive progress in developing MLLMs capable of understanding a more complex and diverse physical world.

## Acknowledgments

This work has been supported in part by the NSFC (No. 62436007), the Key Research and Development Projects in Zhejiang Province (No. 2025C01128, 2024C01106, 2025C01030, 2025C02156), Ningbo Yongjiang Talent Introduction Programme (2023A400-G), Zhejiang University Education Foundation Qizhen Scholar Foundation.

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

## Appendix

In this document, we offer additional details about our benchmark. The appendix is organized as follows:

- § A: Additional details on EOCBench;
- § B: More experimental analysis;
- § C: Experimental setup;
- § D: Additional dataset analysis;
- § E: Limitations and broader impacts;
- § F: Asset license and consent;
- § G: More exemplar visualizations.

## A    Additional Details on EOCBench

### A.1    Human Error Analysis for Evaluation Metrics

To accurately evaluate the Open-Ended Questions ($\mathcal{OQ}$) task, we have developed a novel metric, *Multi-Scale Temporal Accuracy* ($\mathcal{MSTA}$) for comprehensive temporal perception, as introduced in the Section 3.2.4 of the main paper. Here, we provide additional details on the choice of dynamic error margins in $\mathcal{MSTA}$ through the carefully designed human error analysis. Specially, we first asked three volunteers to answer this type of question, and then analyzed the error ratio compared to the ground truth, expressed as $r = (T_{pred} - T_{gt})/T_{gt}$. We compared all $r$ values from all questions and created a histogram of these results, as shown in Fig. 4-(a).

It can be observed that the error ratios are primarily concentrated around 0, with absolute values rarely exceeding 30%. We then conduct a quantile analysis of $|r|$, selecting quantiles at 50%, 75%, 90%, 95%, with corresponding error ratio being 1.4%, 11.9%, 20% and 30%, respectively. Based on these analyses, we set the threshold for dynamic error margins at {1%, 10%, 20%, 30%} for subsequently scale-adaptive boundaries as described in the main paper. A 1% threshold demands near-exact alignment, signifying extreme precision, whereas a 30% threshold caters to greater variability in responses, accommodating almost all human answers within this margin. Fig. 4-(b) depicts the statistical distribution of human error ratio. This dynamic threshold scheme balances strictness and flexibility, ensuring that our framework captures the spectrum of human error while maintaining stringent evaluation criteria.

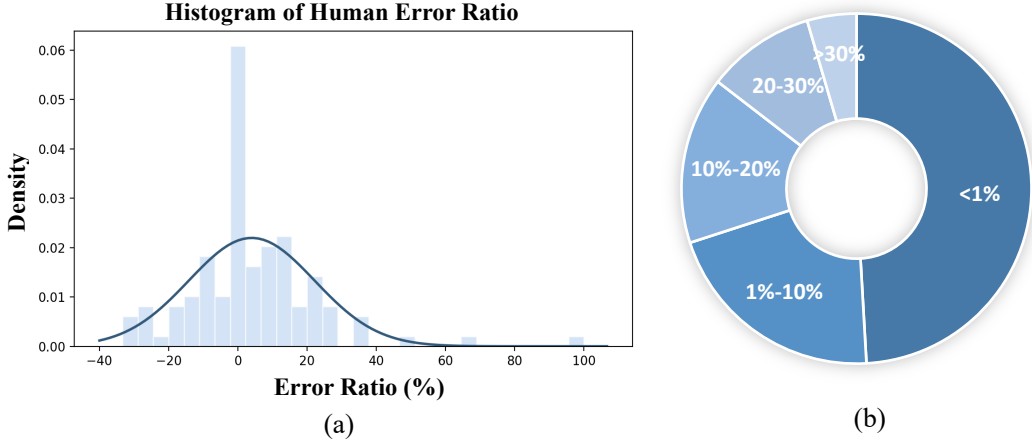

(a)                                                    (b)

Figure 4: **Statistics distribution of human error ratio.** (a) displays a histogram depicting the density and spread of human error ratios; (b) presents a pie chart categorizing the error ratios into quantitative segments.

## A.2 Additional Annotation Details

### A.2.1 Human-in-the-Loop Annotation Pipeline

To ensure the high quality of our benchmark, we employ a rigorous human-in-the-loop annotation pipeline comprising two stages: initial annotation stage, followed by cross-checking and verification stage. Additional details are provided below:

**Initial annotation.** In the initial annotation phase, each annotator is assigned questions belonging to a specific category, such as Past, Present, or Future, along with a set of randomly selected videos to ensure the data diversity and a relatively uniform distribution. To ensure a thorough understanding of each category, annotators are provided with a detailed guide for each category. During the annotation process, annotators are permitted to pause and examine any frame within the video.

**Cross-check and verification.** During the cross-check procedure, each annotator reviews the work of another, focusing primarily on **two key aspects**: the quality of the question-answer pairs and the accuracy of the annotated visual prompts. If a visual prompt, like point, box and mask, is of low quality or missing, the annotator can either carefully reannotate it or discard it; 185 object prompts were reannotated and 34 were discarded. For question-answer pairs that are deemed low quality, the reviewer must definitely discuss the issues with the original annotator to finalize the annotation. In total, 76 questions required collaborative resolution during this verification phase.

Besides, we provide **brief biographies** of all 10 annotators who participated in the annotation process in the Table 5.

| ID | Academic Status | Field of Study |
|----|-----------------|----------------|
| 1 | First-Year Master's Student | Computer Science |
| 2 | First-Year Master's Student | Computer Science |
| 3 | First-Year Master's Student | Robotics |
| 4 | Second-Year Master's Student | Computer Science |
| 5 | Second-Year Master's Student | Computer Science |
| 6 | Second-Year Master's Student | Robotics |
| 7 | Recent Master's Graduate | Robotics |
| 8 | First-Year Ph.D. Student | Computer Science |
| 9 | Third-Year Ph.D. Student | Computer Science |
| 10 | Recent Ph.D. Graduate | Computer Science |

Table 5: Brief biographies of the 10 human annotators in `EOC-Bench`.

# B More Experimental Analysis

## B.1 Comparisons Across Multiple Dimensions

To intuitively showcase the performance of mainstream MLLMs, including both proprietary MLLMs and open-source models, across various evaluation dimensions of `EOC-Bench`, we provide a detailed comparison illustrated in Fig. 5. We assess the models across the 11 evaluation tasks, as well as multiple question types spanning three temporal categories.

## B.2 Analysis of model performance based on video duration

For further analysis, we subcategorize videos into short, medium, and long durations, like [68]. Specifically, short refers to 0–0.5 minutes, medium to 0.5–2 minutes, and long to 2–10 minutes. Across all models, we observe a consistent performance drop as video length increases. For instance, GPT-4o [3] achieves a mean accuracy of 62.57 on short videos, compared to 61.33 on medium and 58.36 on long ones. Other models exhibit similar trends. This trend highlights that most MLLMs demonstrate stronger capabilities in short-term perception and recall, but struggle to maintain memory consistency over extended temporal contexts.

The most significant performance degradation occurs in the Past category, indicating that long-term memory recall remains a critical challenge for current MLLMs in egocentric scenarios. In contrast, Present and Future tasks tend to benefit from medium or long video durations, indicating that extended temporal context helps improving scene understanding and anticipatory reasoning.

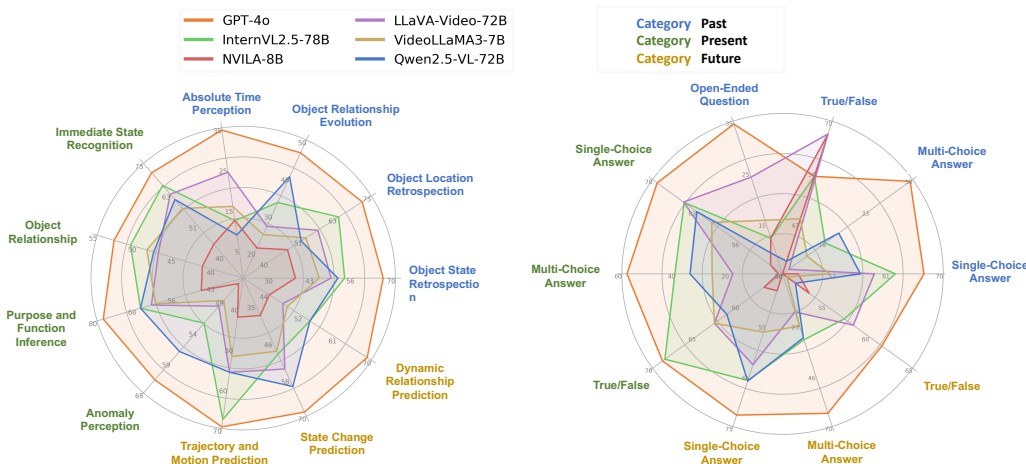

Figure 5: **Comparison of mainstream MLLMs on** `EOC-Bench`. **Left:** Performance on 11 evaluation tasks within `EOC-Bench`. **Right:** Performance across different question types spanning Past, Present and Future categories.

| Model | Input | Mean | | | Past | | | Present | | | Future | | |
|---|---|---|---|---|---|---|---|---|---|---|---|---|---|
| | | Short | Medium | Long | Short | Medium | Long | Short | Medium | Long | Short | Medium | Long |
| GPT-4o [3] | 32f | 62.57 | 61.33 | 58.36 | 57.40 | 52.74 | 49.41 | 66.36 | 69.39 | 75.47 | 65.52 | 75.80 | 68.97 |
| Gemini-2.0-flash [4] | 32f | 57.75 | 56.61 | 57.07 | 50.09 | 43.11 | 48.72 | 65.33 | 67.70 | 67.05 | 53.77 | 71.34 | 67.24 |
| GPT-4o-mini [3] | 32f | 50.17 | 48.25 | 48.21 | 40.10 | 37.07 | 42.19 | 58.51 | 58.08 | 56.82 | 50.34 | 59.24 | 53.45 |
| InternVL2.5-78B [20] | 32f | 52.60 | 52.52 | 50.15 | 44.17 | 37.29 | 38.19 | 60.37 | 65.29 | 64.77 | 50.34 | 70.06 | 65.52 |
| InternVL2.5-38B [20] | 32f | 53.32 | 51.37 | 48.40 | 43.44 | 35.41 | 34.48 | 61.40 | 68.38 | 68.18 | 54.11 | 63.06 | 62.07 |
| Qwen2.5-VL-72B [1] | 1fps | 50.40 | 49.03 | 48.78 | 39.68 | 37.18 | 38.11 | 58.72 | 59.45 | 61.23 | 54.44 | 61.78 | 59.62 |
| LLaVA-Video-72B [65] | 32f | 49.81 | 47.39 | 47.71 | 41.10 | 33.82 | 34.89 | 56.55 | 62.20 | 65.91 | 51.71 | 56.69 | 60.34 |
| LLaVA-OV-72B [27] | 32f | 48.39 | 46.48 | 48.40 | 36.01 | 32.41 | 35.03 | 58.20 | 63.57 | 65.91 | 50.34 | 52.87 | 63.79 |
| VideoLLaMA3-7B [21] | 1fps | 46.41 | 47.31 | 40.32 | 36.99 | 32.47 | 32.01 | 53.87 | 63.92 | 53.41 | 47.96 | 56.69 | 46.55 |
| InternVL2.5-8B [20] | 32f | 42.21 | 41.26 | 43.17 | 33.57 | 32.44 | 35.23 | 49.54 | 49.83 | 51.14 | 41.78 | 48.41 | 55.17 |
| VideoLLaMA2.1-72B [25] | 16f | 41.85 | 41.35 | 40.17 | 33.10 | 31.53 | 28.43 | 46.85 | 51.55 | 54.55 | 49.66 | 49.04 | 55.17 |
| LongVA-7B [31] | 32f | 36.46 | 32.50 | 35.75 | 29.08 | 27.94 | 28.71 | 41.59 | 35.40 | 43.18 | 40.07 | 39.49 | 46.55 |

Table 6: Performance comparison of representative MLLLMs across different video durations.

## B.3 Performance of Various Visual Object Prompts

To validate the effectiveness of visual prompts for object referencing, we conduct additional experiments on the representative MLLMs with various visual prompts, including point, box, and mask. The comparison results are presented in Table 7. In the main paper, we employ box prompt as the default setting. Boxes, compared to masks, are easier to obtain in practical applications. Additionally, compared to points, boxes offer more precise references.

## B.4 Quantitative Error Analysis in EOC-Bench

To quantify and identify the primary challenges of our `EOC-Bench`, we perform a comprehensive error analysis on representative MLLMs, examining both choice-based and open-ended questions.

### B.4.1 Choice-based Questions

For choice-based questions, we conduct analysis on the top-performing MLLM, GPT-4o [3]. Specially, we randomly sampled 300 choice-based erroneous QAs from `EOC-Bench`, with 30 QAs for each task. We then meticulously examined these errors and categorized them into **four primary types**:

- **Perception Error.** This type of error involves issues with perception in the current frame, including interference from previous frames, insufficient attention to finer details, counting errors, and intra-frame interferences.

| Model | Point | | | | Box | | | | Mask | | | |
|---|---|---|---|---|---|---|---|---|---|---|---|---|
| | Mean | Past | Present | Future | Mean | Past | Present | Future | Mean | Past | Present | Future |
| InternVL2.5-8B [20] | 41.83 | 36.35 | 45.48 | 47.51 | 42.05 | 33.45 | 49.70 | 45.36 | 42.12 | 35.04 | 48.07 | 46.15 |
| Qwen2.5VL-7B [1] | 41.21 | 31.81 | 49.18 | 46.35 | **43.13** | 31.38 | 53.93 | 47.35 | 41.31 | 30.64 | 52.08 | 42.60 |
| VideoLLaMA3-7B [21] | 45.24 | 37.04 | 52.89 | 47.93 | 46.04 | 35.00 | 56.01 | 50.49 | **46.10** | 35.41 | 55.93 | 49.94 |
| LLaVA-Video-7B [65] | **41.13** | 32.82 | 49.04 | 43.39 | 39.50 | 32.67 | 48.96 | 43.39 | 40.91 | 32.40 | 48.96 | 43.39 |

Table 7: Performance of representative MLLMs with different visual prompt inputs.

- **Memory Error.** This error type reflects incorrect observation or recall of information from previous frames, including interference from current frames and missing observations, suggesting that the 32 sampled frames are insufficient to answer the memory-related questions.

- **Relational Reasoning Error.** This type of error involves difficulties in perceiving or inferring simple relationships between objects.

- **Knowledge Error.** This category encompasses errors in reasoning, common sense, and calculation.

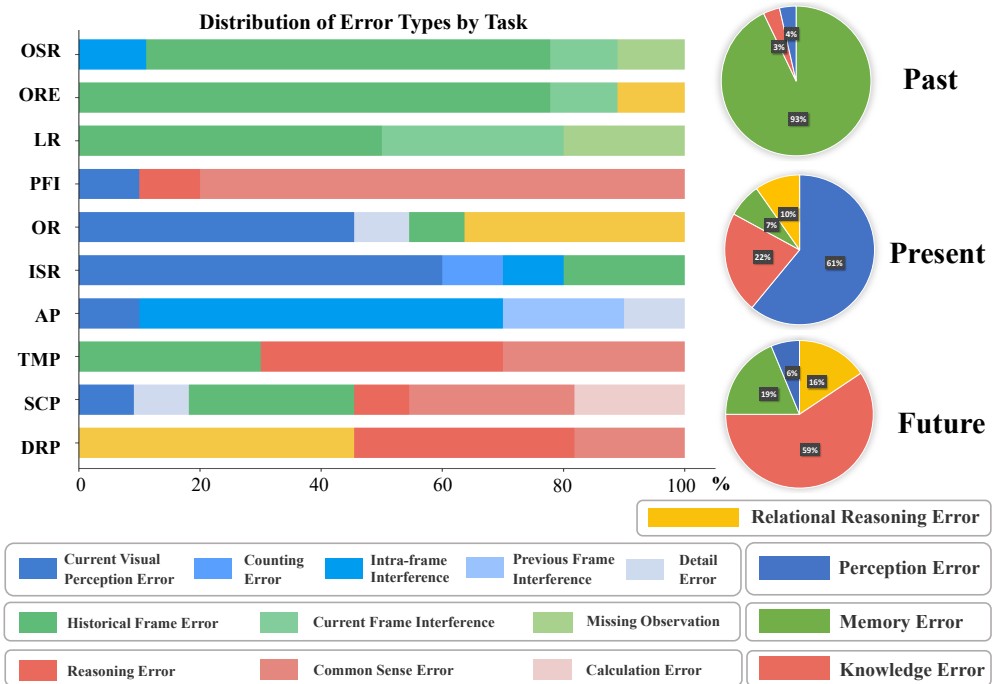

Figure 6: Quantitative error analysis by type for choice-based questions in `EOC-Bench`.

In the **Past** category, as illustrated in Fig. 6, memory errors are predominant, accounting for 93% of the errors. These are primarily due to insufficient processing of historical frames (73%) and interference from current frame (17%). The remaining 10% are missing observation errors, which highlight the inherent constraints of fixed-frame sampling strategies. These findings point to a significant weakness of GPT-4o [3] in temporal context modeling, particularly its difficulty in effectively retaining and using cross-frame information for video understanding tasks.

In the **Present** category, perception errors account for 61%, followed by knowledge errors (22%) and memory errors (7%). Notably, intra-frame interference constitutes a significant portion of perception errors, revealing the model's limitations in regional-level visual perception and its susceptibility to hallucinatory artifacts. These observations suggest that spatial perception remains a persistent challenge.

In the **Future** category, approximately 59% of errors are knowledge-related issues, indicating limitations in reasoning abilities and common sense understanding.

### B.4.2    Open-Ended Questions

To assess open-ended questions related to temporal perception accuracy, we conducted a density-based analysis of deviations between ground-truth timestamps and model-generated responses, as

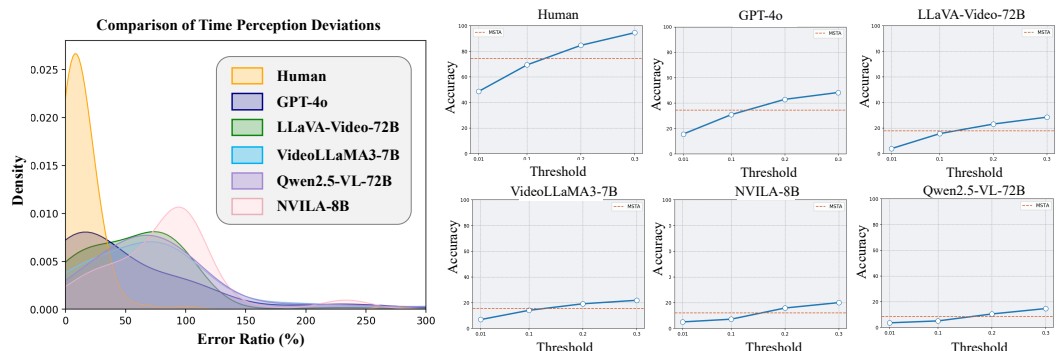

Figure 7: Quantitative error analysis for open-ended questions in `EOC-Bench`. **Left:** Density analysis of temporal perception deviations (error ratio) among humans and models. **Right:** Model accuracy across different time thresholds for dynamic error margins.

visualized in Fig. 7-(Left). The distribution of human responses exhibits a pronounced peak followed by rapid decay, suggesting that most human answers achieve minimal error ratios, with only sporadic instances of higher inaccuracies. In contrast, the five top-performing models–GPT-4o [3], LLaVA-Video-72B [65], VideoLLaMA3-7B [21], Qwen2.5-VL-72B [1] and NVILA-8B [66]–demonstrate flatter distributions with broader spreads. This pattern suggests that these models exhibit greater variability in temporal perception, frequently producing larger errors in specific cases.

The observed disparity highlights a substantial discrepancy between current MLLMs and human-level temporal perception, suggesting that some model predictions rely on haphazard estimation rather than precise temporal understanding. As illustrated in Fig. 7-(Right), the figure also presents model accuracy across various time thresholds, specifically 0.01, 0.1, 0.2, and 0.3.

## C  Experimental Setup

### C.1  Model Configurations

The configurations of the mainstream MLLMs we evaluate, including the official checkpoints, the number of frame samples, and details regarding the "Do Sample", "Max New Tokens", "Temperature" and "Top-P" parameters, are provided in Table 8.

### C.2  Additional Implementation Details

We utilize the official repository of each MLLMs to perform evaluations on our `EOC-Bench` benchmark. Pre-sampled images from 1-frame, 8-frame, 16-frame, 32-frame, and 1 fps sequences serve as input for the corresponding models according to their default settings. Besides, for proprietary MLLMs, including GPT-4o, GPT-4o-mini and Gemini, we introduce timestamps prior to each frame to enhance the model's temporal awareness. The open-source models are evaluated with their default settings, and all evaluations are conducted using NVIDIA A100 GPUs.

### C.3  Carefully Crafted Prompts

**Visual Prompts.** We employ the SoM [69] method to overlay various spatial markers onto the images in the final frame of the video. For a single object, only its visual prompt is highlighted in red on the last frame. In the case of multiple objects, we overlay both their identifying numbers and visual prompts in various colors to facilitate differentiation. An example is presented in Fig. 8.

**Text Prompts.** The text prompts used for inference are consistent across all models and are as follows:

```
System Prompt: I have overlaid the box on the last frame of the video, <
object 1>: red; <object 2>: blue, <object 3>: green; <object 4>:
yellow; <object 5>: purple; <object 6>: orange;
```

| Model | Frames | API Checkpoint / HF Checkpoint | Do Sample | Max New Tokens | Temp. | Top-P |
|---|---|---|---|---|---|---|
| *Proprietary Multimodal Foundation Models* | | | | | | |
| GPT-4o-mini [3] | 32 | gpt-4o-mini-2024-07-18 | | 1024 | 0 | 1 |
| GPT-4o [3] | 32 | gpt-4o-2024-08-06 | | 1024 | 0 | 1 |
| Gemini-2.0-Flash [4] | 32 | gemini-2.0-flash | | 1024 | 0 | 1 |
| *Open-Source Multimodal Foundation Models* | | | | | | |
| InternVL2.5-8B [20] | 32 | OpenGVLab/InternVL2_5-8B | False | 1024 | | |
| InternVL2.5-38B [20] | 32 | OpenGVLab/InternVL2_5-38B | False | 1024 | | |
| InternVL2.5-78B [20] | 32 | OpenGVLab/InternVL2_5-78B | False | 1024 | | |
| LongVA-7B [31] | 32 | lmms-lab/LongVA-7B | False | 1024 | | |
| LLaVA-Video-7B [65] | 32 | lmms-lab/LLaVA-Video-7B-Qwen2 | False | 1024 | | |
| LLaVA-Video-72B [65] | 32 | lmms-lab/LLaVA-Video-72B-Qwen2 | False | 1024 | | |
| LLaVA-OneVision-7B [27] | 32 | lmms-lab/llava-onevision-qwen2-7b-ov | False | 1024 | | |
| LLaVA-OneVision-72B [27] | 32 | lmms-lab/llava-onevision-qwen2-72b-ov-sft | False | 1024 | | |
| Qwen2.5-VL-3B [1] | 1fps | Qwen/Qwen2.5-VL-3B-Instruct | False | 1024 | | |
| Qwen2.5-VL-7B [1] | 1fps | Qwen/Qwen2.5-VL-7B-Instruct | False | 1024 | | |
| Qwen2.5-VL-72B [1] | 1fps | Qwen/Qwen2.5-VL-72B-Instruct | False | 1024 | | |
| VideoLLaMA2.1-7B [25] | 16 | DAMO-NLP-SG/VideoLLaMA2.1-7B | False | 1024 | | |
| VideoLLaMA2-72B [25] | 32 | DAMO-NLP-SG/VideoLLaMA2-72B | False | 1024 | | |
| VideoLLaMA3-2B [21] | 1fps | DAMO-NLP-SG/VideoLLaMA3-2B | False | 1024 | | |
| VideoLLaMA3-7B [21] | 1fps | DAMO-NLP-SG/VideoLLaMA3-7B | False | 1024 | | |
| NVILA-8B [66] | 32 | Efficient-Large-Model/NVILA-8B-Video | False | 1024 | | |
| VideoLLaVA-7B [28] | 8 | LanguageBind/Video-LLaVA-7B | False | 1024 | | |
| VideoRefer-7B [12] | 16 | DAMO-NLP-SG/VideoRefer-7B | False | 1024 | | |
| ViP-LLaVA-7B [23] | 1 | llava-hf/vip-llava-7b-hf | False | 1024 | | |
| Osprey-7B [22] | 1 | sunshine-lwt/Osprey-Chat-7b | False | 1024 | | |
| SPHINX-V-13B [67] | 1 | Afeng-x/SPHINX-V-Model | False | 1024 | | |

Table 8: Model configurations for evaluating mainstream MLLMs in EOCBench (Temp.: temperature).

```
Single choice
USER: {Question} Options: {Options} Answer directly using the letters of
the options given and wrap your response in <choice></choice>. For
example, if the answer is A, then output <choice>A</choice>.

Multi choice
USER: {Question} Options: {Options} Answer directly using the letters of
the options given. There are multiple answers, so wrap your response
in <choice></choice>. For example, if the answer is A and B, then
output <choice>A, B</choice>; if the answer is A, B and C, then output
 <choice>A, B, C</choice>.

Open ended
USER: {Question} Please output the answer directly in seconds.
```

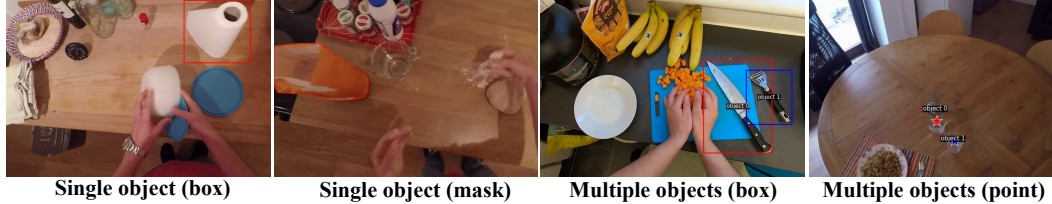

| Single object (box) | Single object (mask) | Multiple objects (box) | Multiple objects (point) |
|---|---|---|---|

Figure 8: Illustrative examples of annotation formats for visual prompts.

# D  Additional Dataset Analysis

Fig. 9 provides an additional statistical analysis through the form **word cloud**, capturing the range of questions and answers encompassed in EOC-Bench. The predominance of terms related to dynamics and changes—such as "**change**", "**changed**", "**seconds**", and "**happened**"—indicates a substantial focus on the temporal and transformational aspects within the dataset. These terms are essential for assessing the memory capabilities of robots, as they require understanding and recalling sequences of events and alterations over time.

Moreover, the word cloud highlights the significance of spatial understanding, with frequent terms like "**relative**", "**relative position**", and "**relationship**". These words underscore the importance of comprehending the spatial dynamics between objects. Analyzing these spatial relationships allows the model to infer how objects are positioned relative to each other, providing insights essential for effective planning and execution.

## E    Limitations and Broader Impacts

**Limitations.**  Our EOC-Bench demonstrates the commendable assessment of embodied object cognition, yet certain limitation remain. The video inputs for EOC-Bench are limited to durations of under six minutes, which may not adequately evaluate the cognitive abilities of MLLMs in terms of prolonged visual memory.

In our future work, we are dedicated to progressing our research by collecting video resources of longer durations. We aims to explore the effects of increasing video input durations, particularly in terms of the models' ability to retain prolonged visual information.

**Broader Impacts.** As a benchmark specifically designed for evaluating in the domain of embodied ego-centric cognition, EOC-Bench is set to draw considerable interest from researchers keen on examining cognitive processes related to focusing on specific objects. Moreover, EOC-Bench aims to assist contemporary MLLMs in transcending the limitations inherent in images, videos, and texts alone, by shifting their focus toward the visual prompt inputs encountered in real-world scenarios.

## F    Asset License and Consent

In our  EOC-Bench, we utilize four open-source datasets: EPIC-KITCHENS [62], Ego4D [60], Charades-ego [63] and MECCANO [64]. All datasets are publicly accessible and freely available for academic research. Table 9 provides a detailed list of the resources used in this research work, along with their respective licenses.

| Dataset | License | URL |
|---|---|---|
| EPIC-KITCHENS [62] | CC BY-NC 4.0 | https://epic-kitchens.github.io/2025 |
| Ego4D [60] | MIT license | https://github.com/facebookresearch/Ego4d |
| Charades-ego [63] | Non-Commercial Use | https://prior.allenai.org/projects/charades-ego |
| MECCANO [64] | CC BY-NC 4.0 | https://iplab.dmi.unict.it/MECCANO/ |

Table 9: Open-source resources used in this work.

## G    More Exemplar Visualizations

### G.1    Failure Case Studies

Fig. 10 displays representative cases from top-performing GPT-4o [3] on our EOC-Bench. These cases systematically demonstrate the model's failure patterns across multiple error categories, including current visual perception errors, common sense errors, and historical frame errors, while covering diverse question types from EOC-Bench.

### G.2    Visual Samples Across Tasks

To intuitively illustrate the characteristics of our EOC-Bench, we further showcase samples spanning 11 tasks, organized as follows:

- Object State Retrospection (Fig. 11)
- Object Location Retrospection (Fig. 12)
- Object Relationship Evolution (Fig. 13)
- Absolute Time Perception (Fig. 14)
- Immediate State Recognition (Fig. 15)
- Object Relationship (Fig. 16)

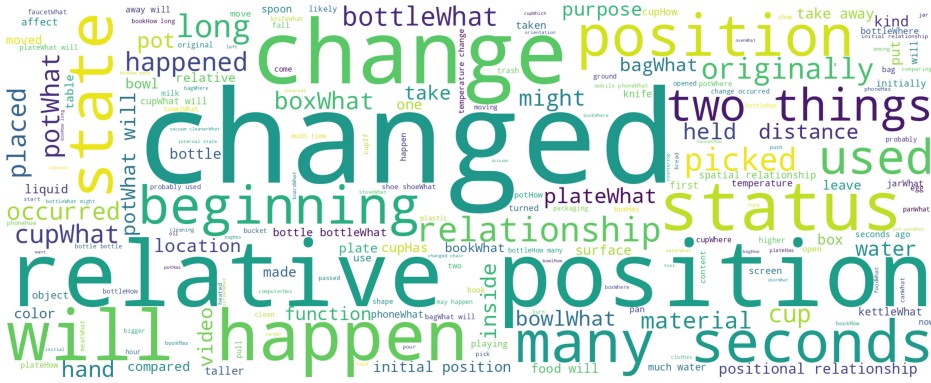

Figure 9: Worldcloud of questions and answers in EOC-Bench.

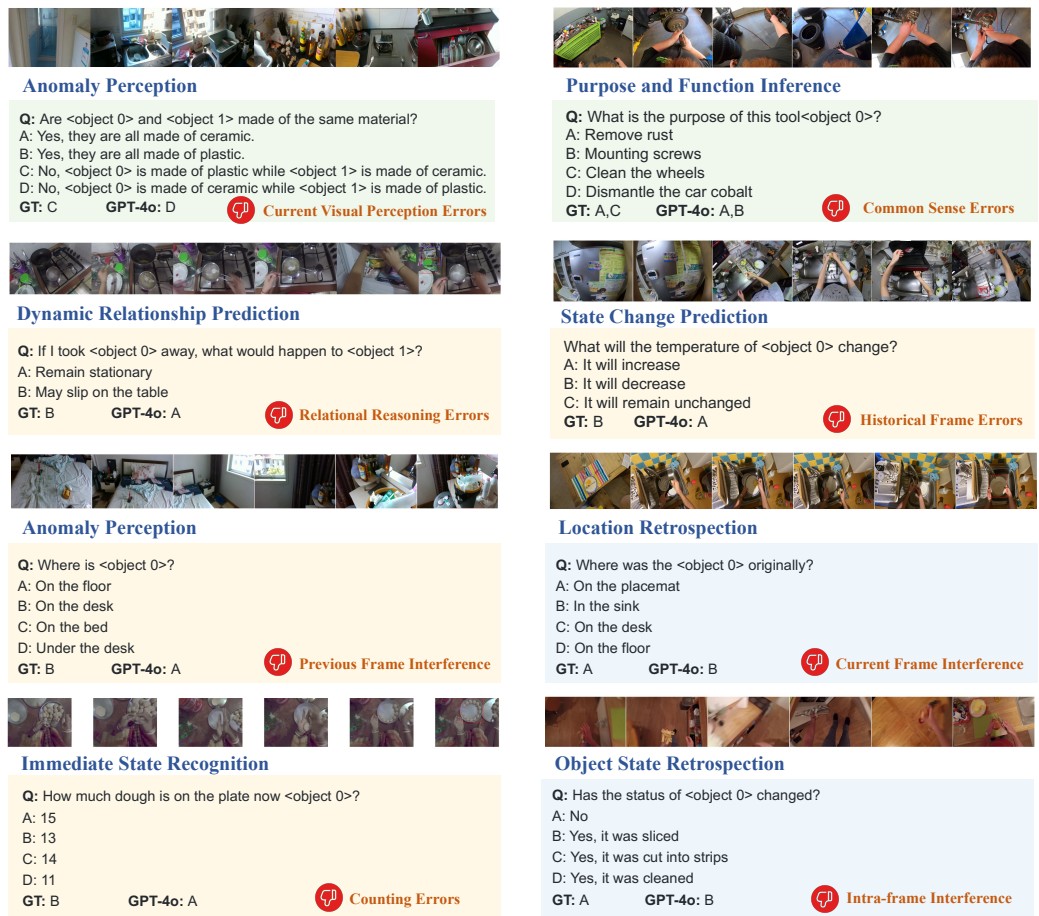

Figure 10: Failure cases of the top-performing GPT-4o on EOC-Bench.

- Purpose and Function Inference (Fig. 17)
- Anomaly Perception (Fig. 18)
- Trajectory and Motion Prediction (Fig. 19)
- State Change Prediction (Fig. 20)
- Dynamic Relationship Prediction (Fig. 21)

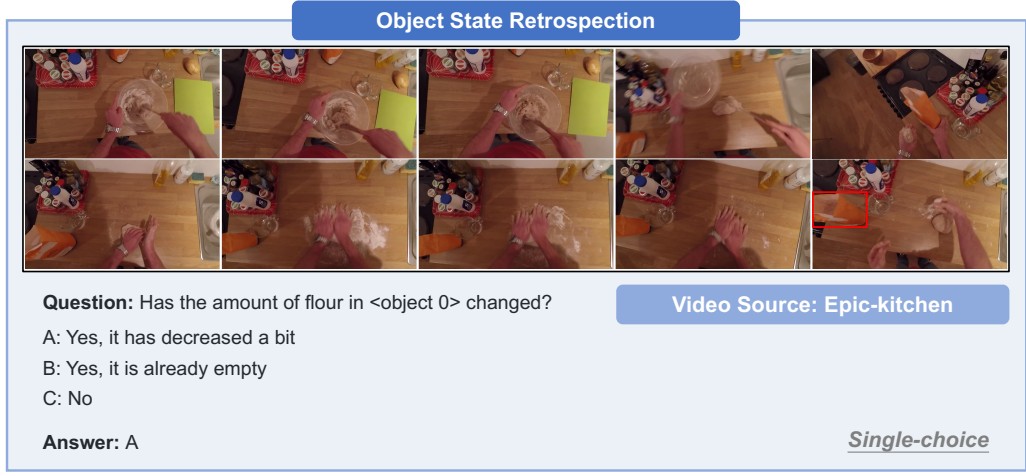

**Object State Retrospection**

**Question:** Has the amount of flour in <object 0> changed?

A: Yes, it has decreased a bit

B: Yes, it is already empty

C: No

**Video Source: Epic-kitchen**

**Answer:** A

*Single-choice*

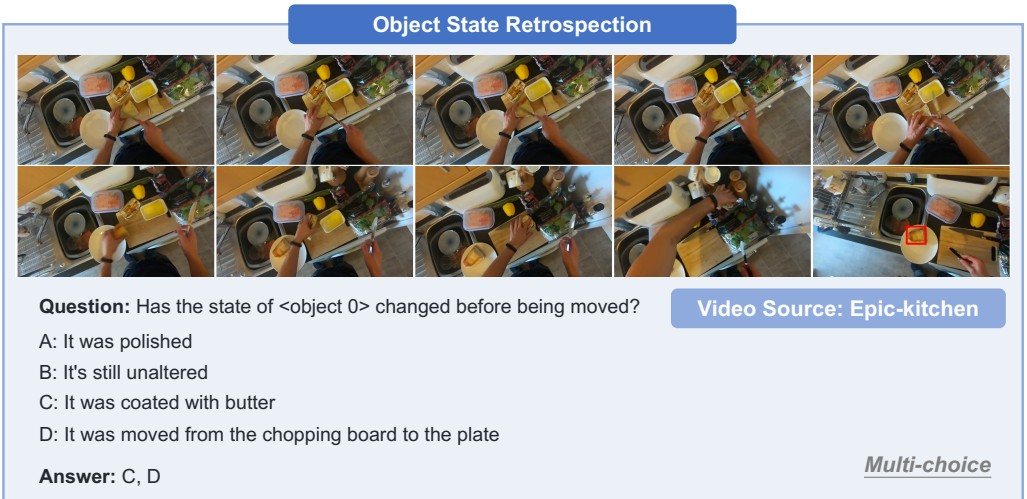

**Object State Retrospection**

**Question:** Has the state of <object 0> changed before being moved?

A: It was polished

B: It's still unaltered

C: It was coated with butter

D: It was moved from the chopping board to the plate

**Video Source: Epic-kitchen**

**Answer:** C, D

*Multi-choice*

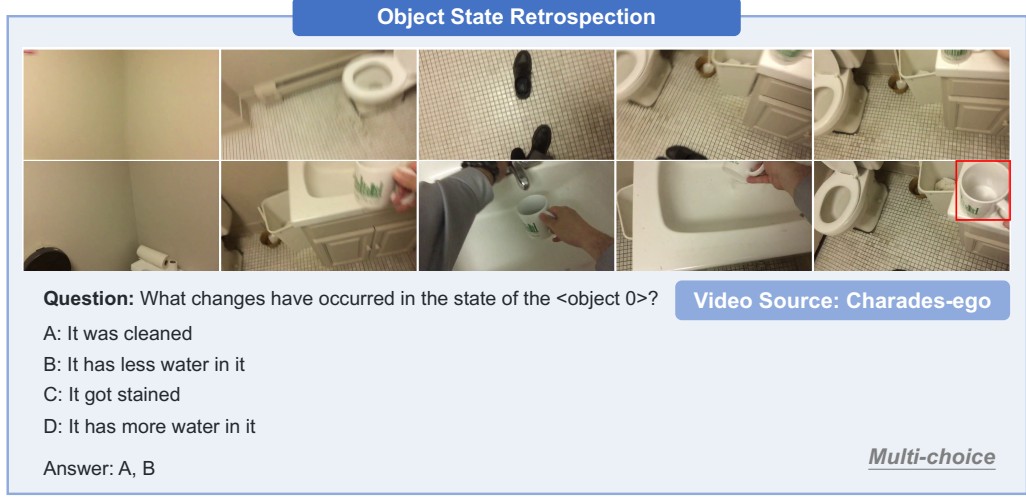

**Object State Retrospection**

**Question:** What changes have occurred in the state of the <object 0>?

A: It was cleaned

B: It has less water in it

C: It got stained

D: It has more water in it

**Video Source: Charades-ego**

Answer: A, B

*Multi-choice*

Figure 11: Visualization of samples in **Object State Retrospection (Past)**.

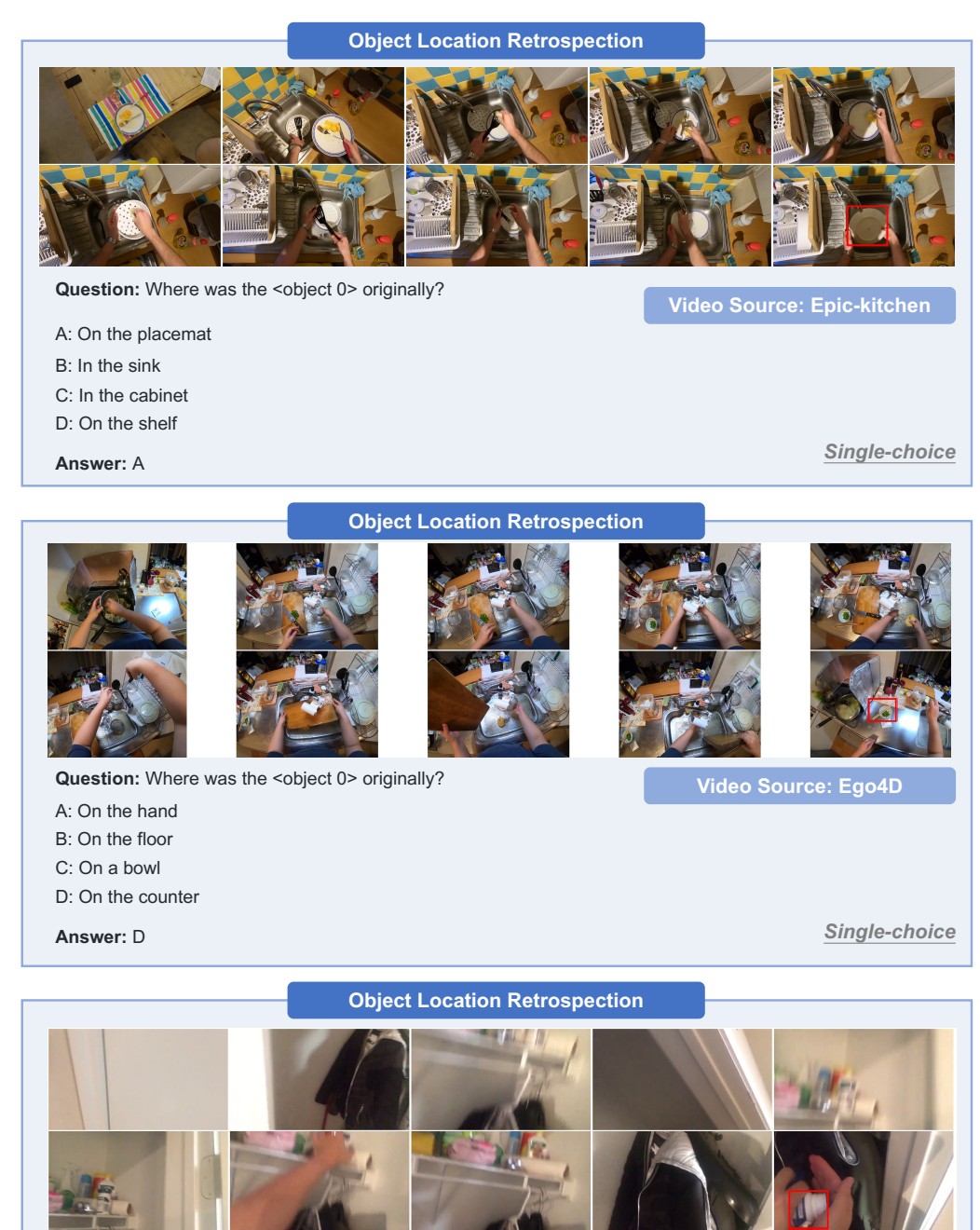

**Object Location Retrospection**

**Question:** Where was the <object 0> originally?

**Video Source: Epic-kitchen**

A: On the placemat
B: In the sink
C: In the cabinet
D: On the shelf

*Single-choice*

**Answer:** A

**Object Location Retrospection**

**Question:** Where was the <object 0> originally?

**Video Source: Ego4D**

A: On the hand
B: On the floor
C: On a bowl
D: On the counter

*Single-choice*

**Answer:** D

**Object Location Retrospection**

**Question:** What was the initial position of the <object 0>?

**Video Source: Charades-ego**

A: On the shelf
B: In hand
C: In the wardrobe
D: On the floor

*Multi-choice*

**Answer:** A,C

Figure 12: Visualization of samples in **Location Retrospection (Past)**.

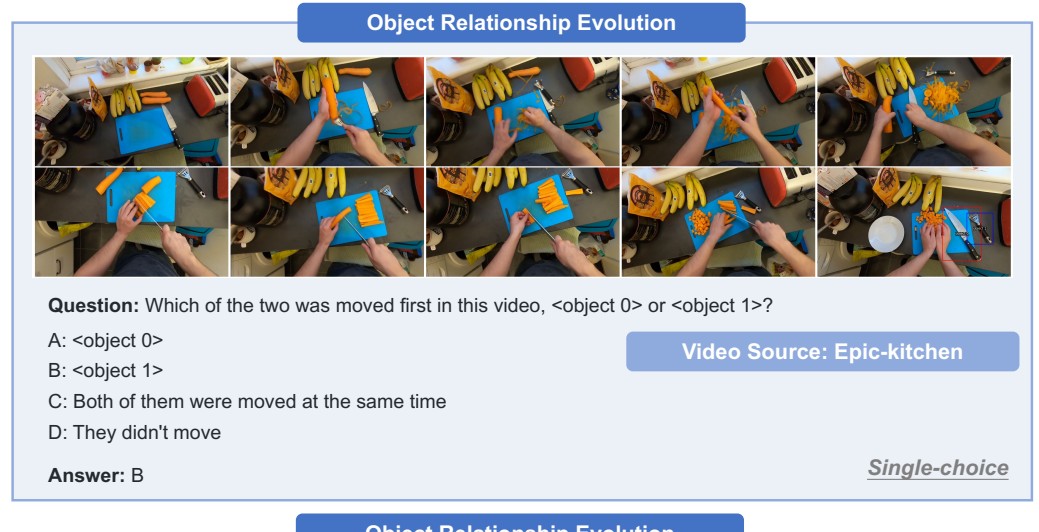

**Question:** Which of the two was moved first in this video, <object 0> or <object 1>?

A: <object 0>

B: <object 1>

C: Both of them were moved at the same time

D: They didn't move

**Video Source: Epic-kitchen**

**Answer:** B

*Single-choice*

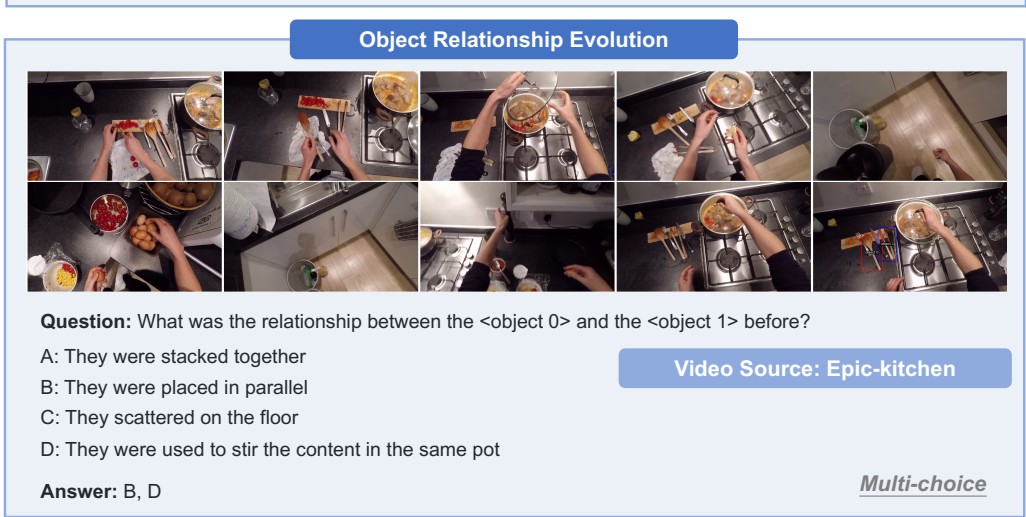

**Question:** What was the relationship between the <object 0> and the <object 1> before?

A: They were stacked together

B: They were placed in parallel

C: They scattered on the floor

D: They were used to stir the content in the same pot

**Video Source: Epic-kitchen**

**Answer:** B, D

*Multi-choice*

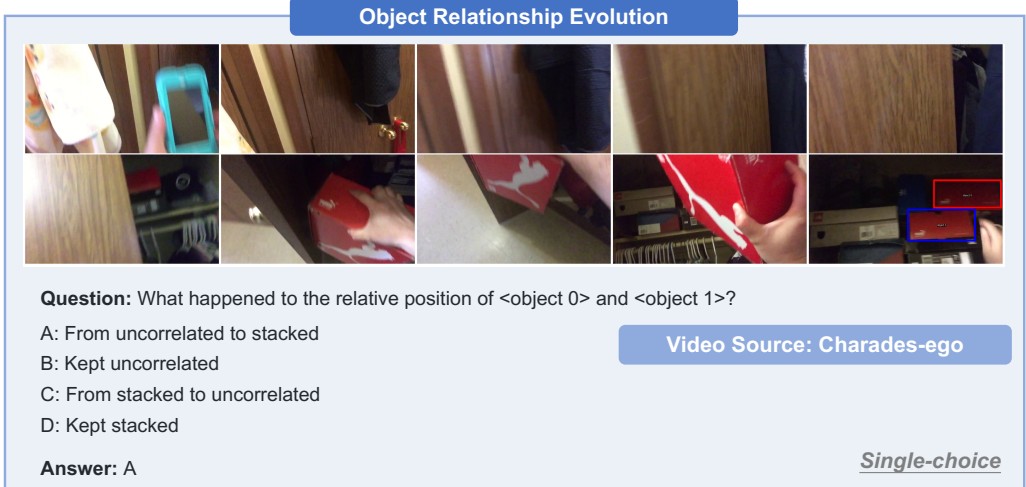

**Question:** What happened to the relative position of <object 0> and <object 1>?

A: From uncorrelated to stacked

B: Kept uncorrelated

C: From stacked to uncorrelated

D: Kept stacked

**Video Source: Charades-ego**

**Answer:** A

*Single-choice*

Figure 13: Visualization of samples in **Object Relationship Evolution (Past)**.

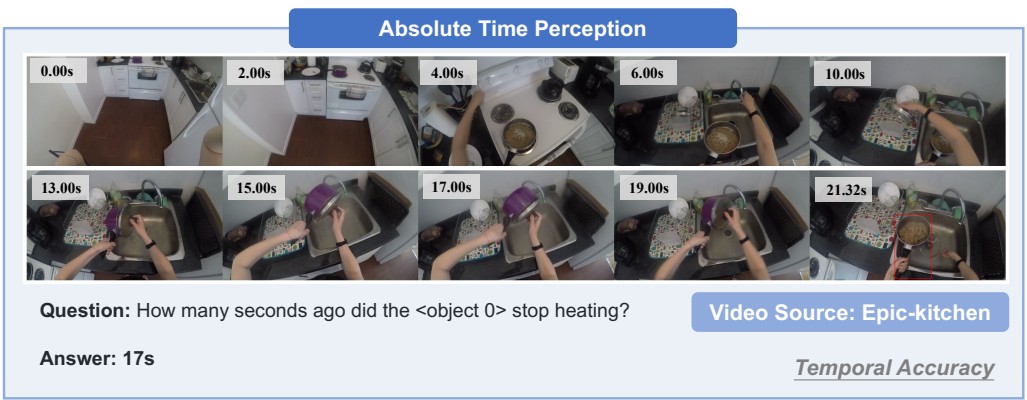

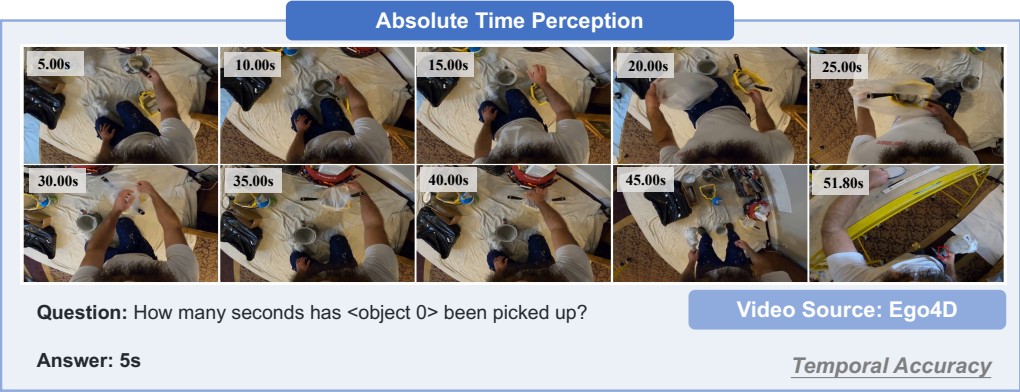

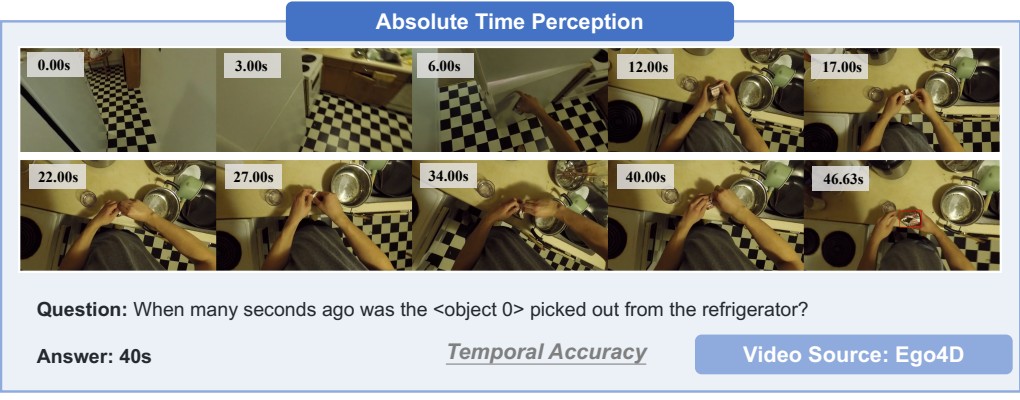

Figure 14: Visualization of samples in **Absolute Time Perception (Past)**.

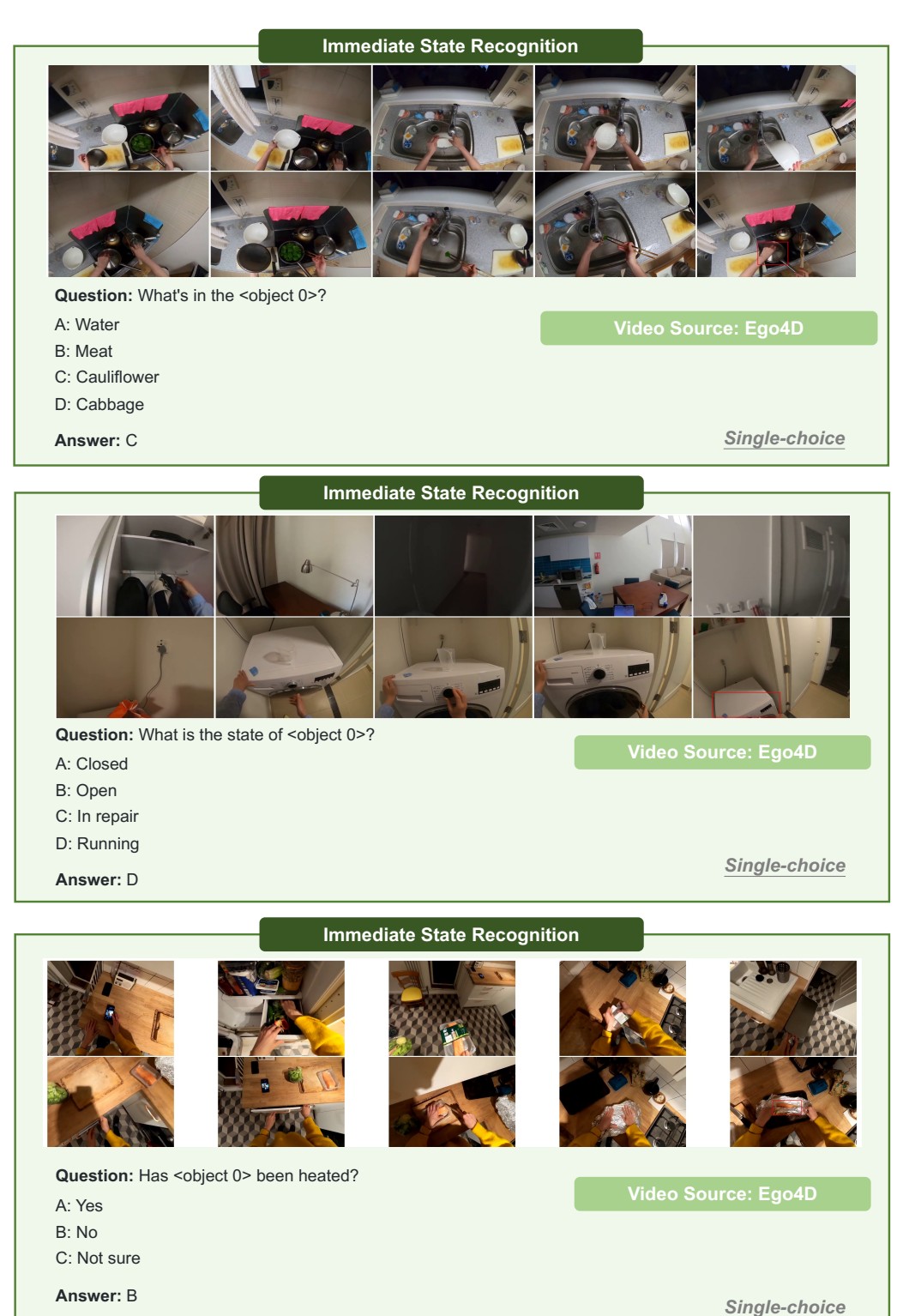

Figure 15: Visualization of samples in **Immediate State Recognition (Present)**.

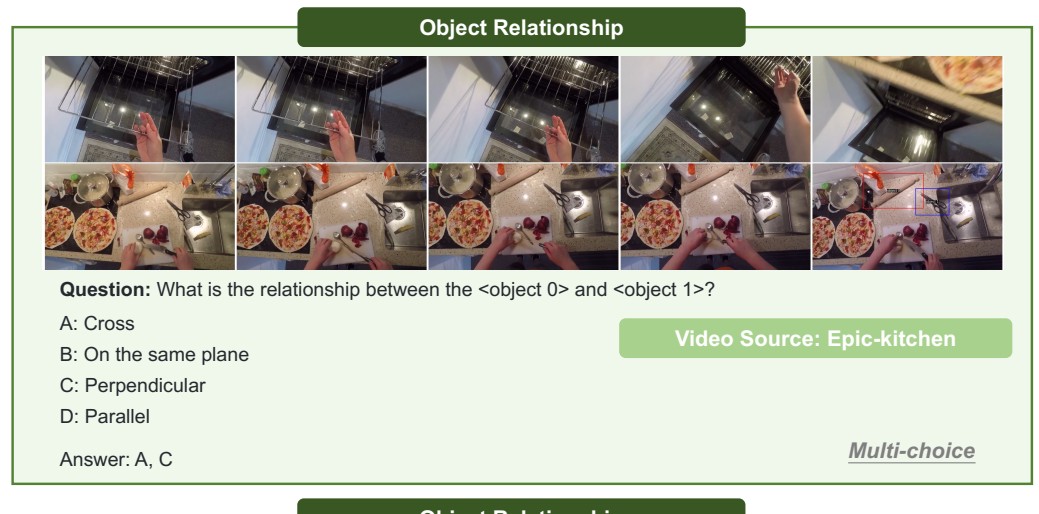

**Object Relationship**

**Question:** What is the relationship between the <object 0> and <object 1>?

A: Cross

B: On the same plane

C: Perpendicular

D: Parallel

Video Source: Epic-kitchen

Answer: A, C

*Multi-choice*

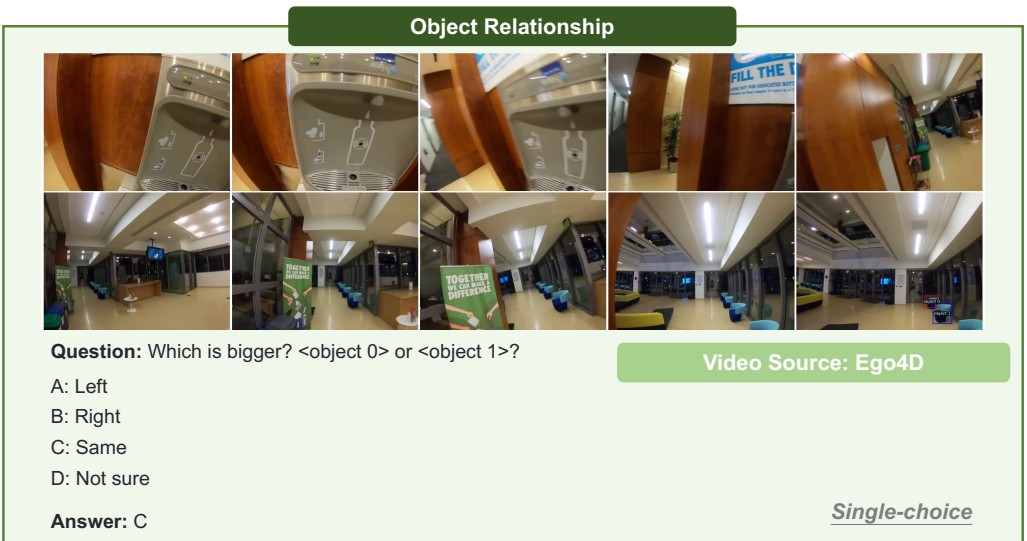

**Object Relationship**

**Question:** Which is bigger? <object 0> or <object 1>?

A: Left

B: Right

C: Same

D: Not sure

Video Source: Ego4D

**Answer:** C

*Single-choice*

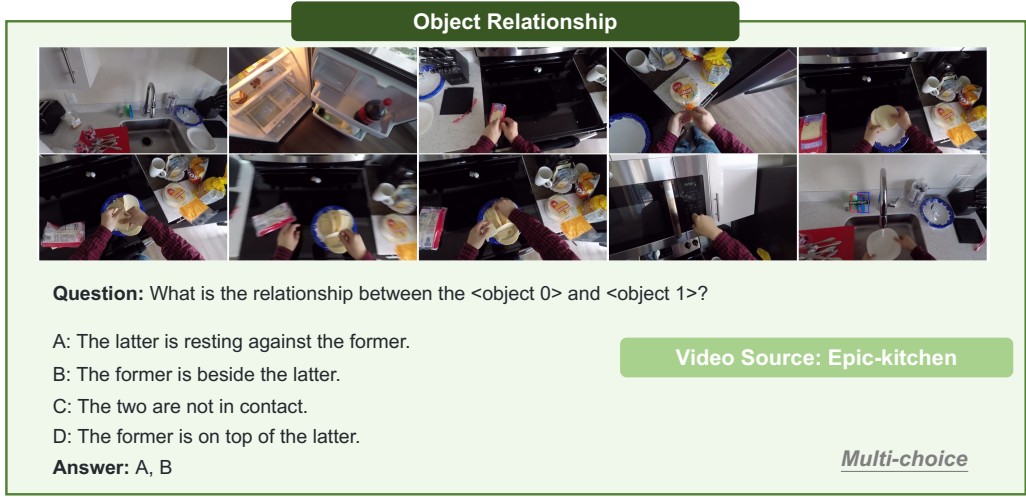

**Object Relationship**

**Question:** What is the relationship between the <object 0> and <object 1>?

A: The latter is resting against the former.

B: The former is beside the latter.

C: The two are not in contact.

D: The former is on top of the latter.

Video Source: Epic-kitchen

**Answer:** A, B

*Multi-choice*

Figure 16: Visualization of samples in **Object Relationship (Present)**.

## Purpose and Function Inference

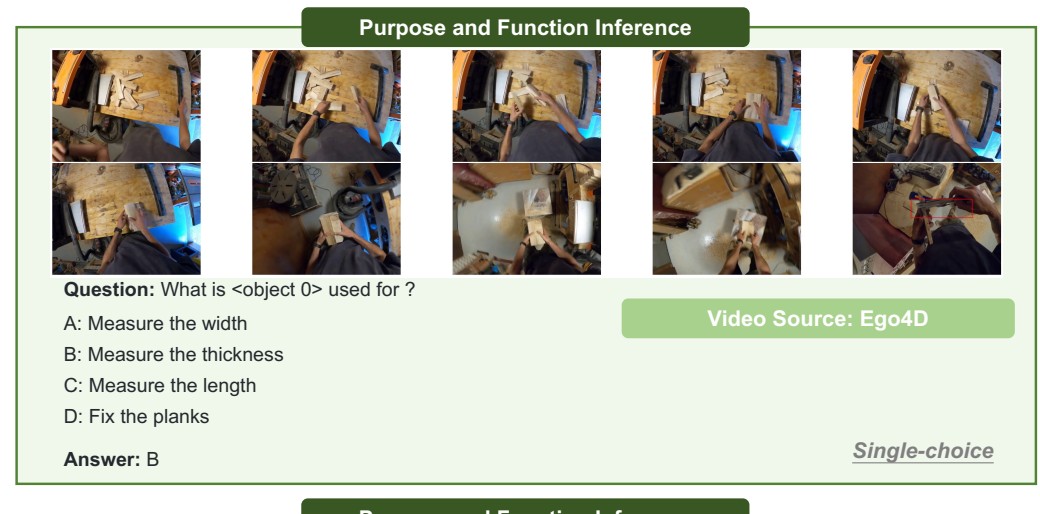

**Question:** What is <object 0> used for ?

A: Measure the width

B: Measure the thickness

C: Measure the length

D: Fix the planks

Video Source: Ego4D

**Answer:** B

*Single-choice*

## Purpose and Function Inference

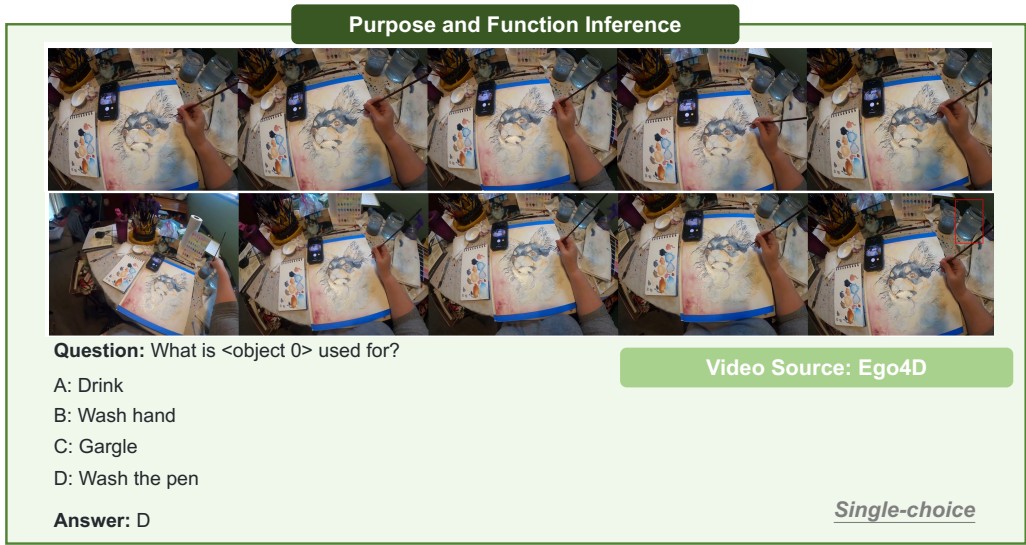

**Question:** What is <object 0> used for?

A: Drink

B: Wash hand

C: Gargle

D: Wash the pen

Video Source: Ego4D

**Answer:** D

*Single-choice*

## Purpose and Function Inference

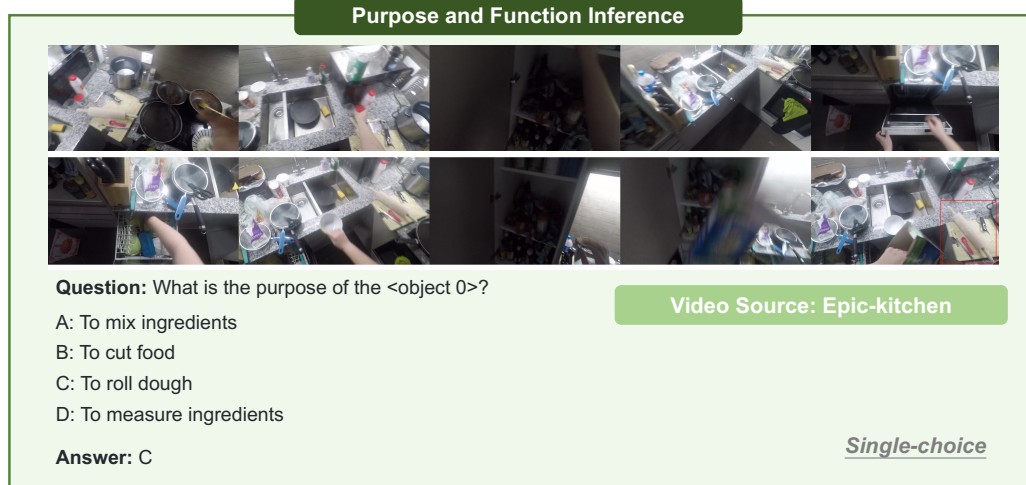

**Question:** What is the purpose of the <object 0>?

A: To mix ingredients

B: To cut food

C: To roll dough

D: To measure ingredients

Video Source: Epic-kitchen

**Answer:** C

*Single-choice*

Figure 17: Visualization of samples in **Purpose and Function Inference (Present)**.

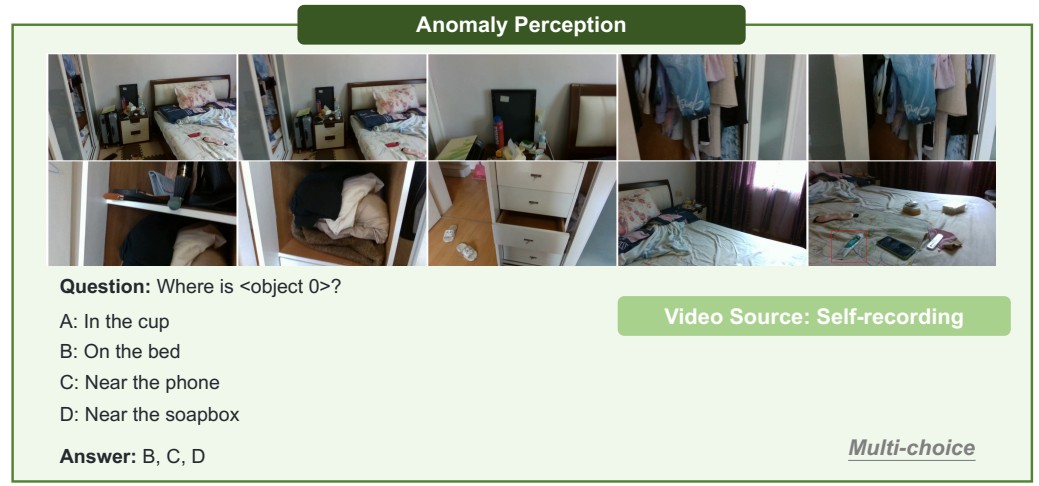

**Question:** Where is <object 0>?

A: In the cup

B: On the bed

C: Near the phone

D: Near the soapbox

**Video Source: Self-recording**

**Answer:** B, C, D

*Multi-choice*

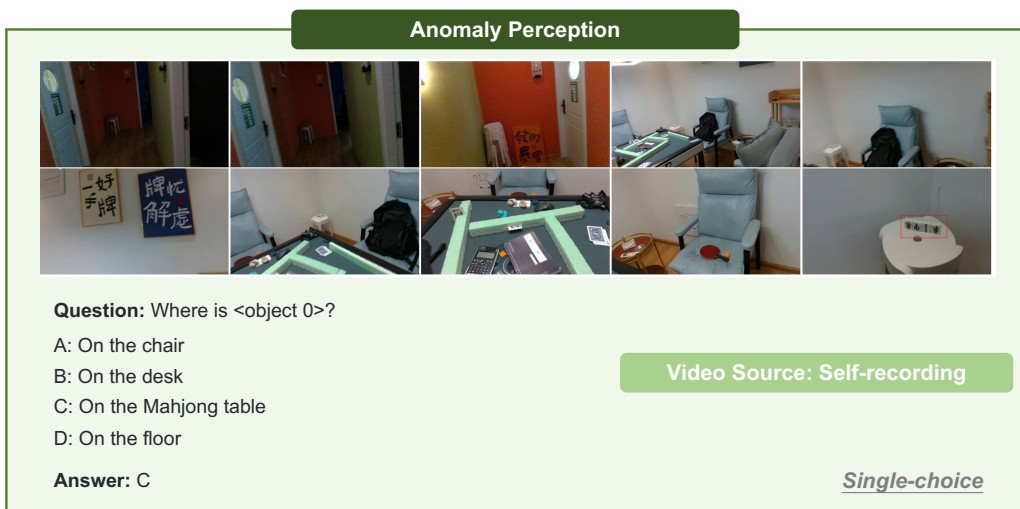

**Question:** Where is <object 0>?

A: On the chair

B: On the desk

C: On the Mahjong table

D: On the floor

**Video Source: Self-recording**

**Answer:** C

*Single-choice*

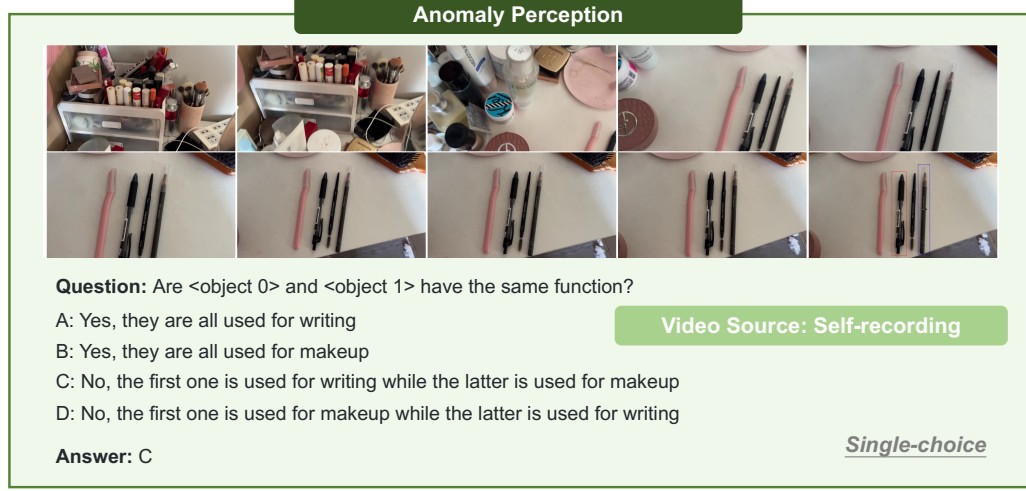

**Question:** Are <object 0> and <object 1> have the same function?

A: Yes, they are all used for writing

B: Yes, they are all used for makeup

C: No, the first one is used for writing while the latter is used for makeup

D: No, the first one is used for makeup while the latter is used for writing

**Video Source: Self-recording**

**Answer:** C

*Single-choice*

Figure 18: Visualization of samples in **Anomaly Perception (Present)**.

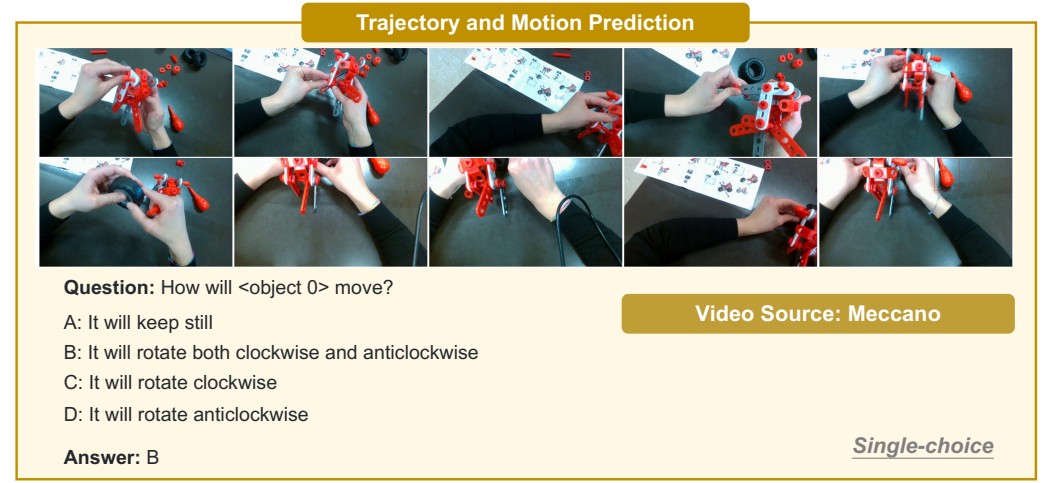

**Trajectory and Motion Prediction**

**Question:** How will <object 0> move?

A: It will keep still

B: It will rotate both clockwise and anticlockwise

C: It will rotate clockwise

D: It will rotate anticlockwise

**Video Source: Meccano**

**Answer:** B

*Single-choice*

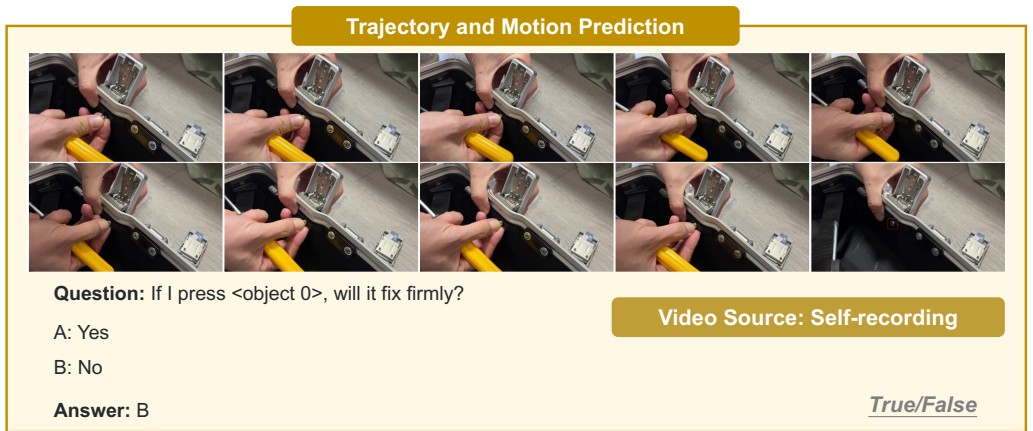

**Trajectory and Motion Prediction**

**Question:** If I press <object 0>, will it fix firmly?

A: Yes

B: No

**Video Source: Self-recording**

**Answer:** B

*True/False*

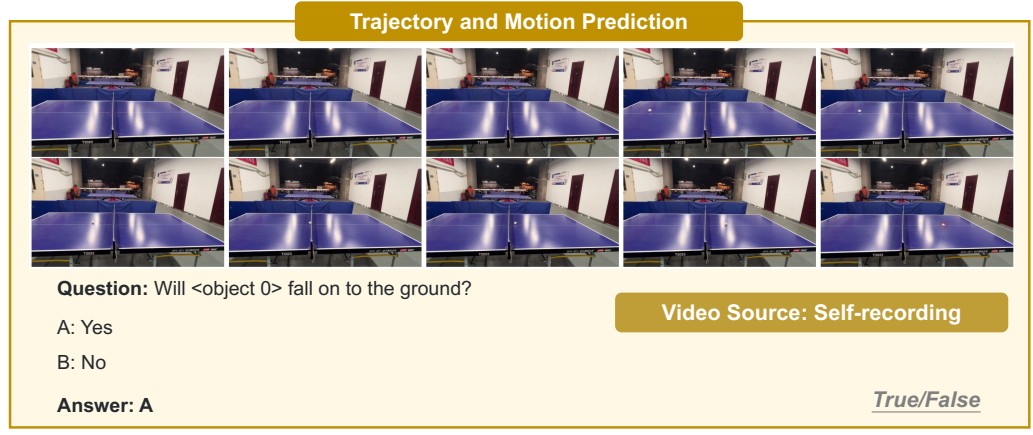

**Trajectory and Motion Prediction**

**Question:** Will <object 0> fall on to the ground?

A: Yes

B: No

**Video Source: Self-recording**

**Answer:** A

*True/False*

Figure 19: Visualization of samples in **Trajectory and Motion Prediction (Future)**.

**State Change Prediction**

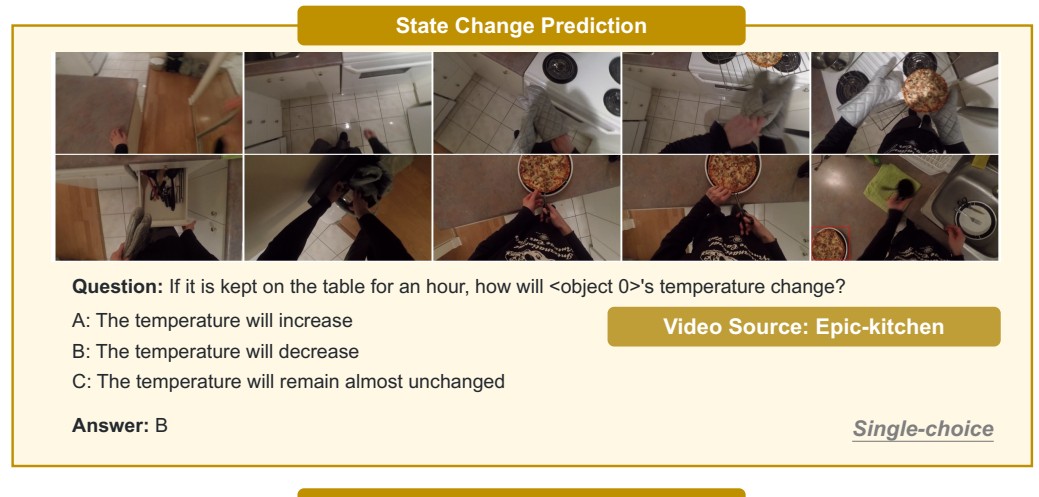

**Question:** If it is kept on the table for an hour, how will <object 0>'s temperature change?

A: The temperature will increase

B: The temperature will decrease

C: The temperature will remain almost unchanged

**Video Source: Epic-kitchen**

**Answer:** B

*Single-choice*

---

**State Change Prediction**

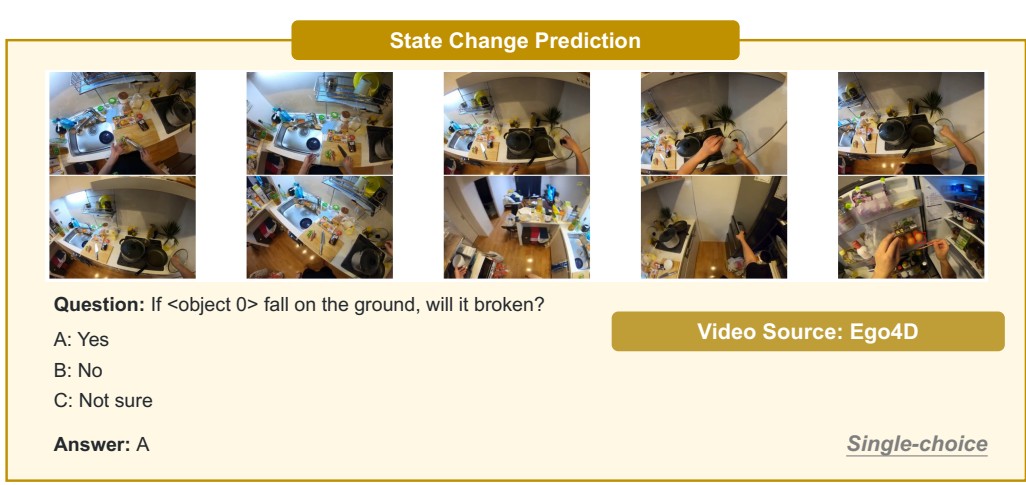

**Question:** If <object 0> fall on the ground, will it broken?

A: Yes

B: No

C: Not sure

**Video Source: Ego4D**

**Answer:** A

*Single-choice*

---

**State Change Prediction**

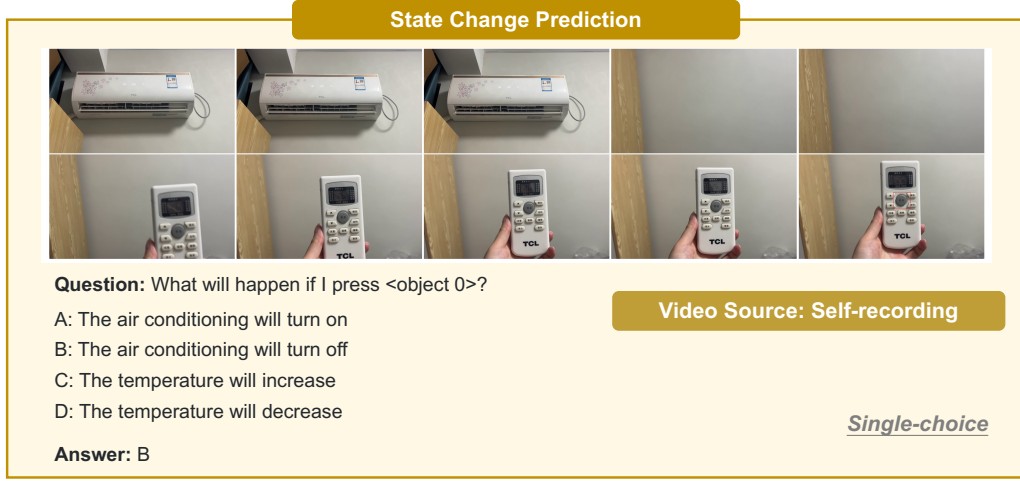

**Question:** What will happen if I press <object 0>?

A: The air conditioning will turn on

B: The air conditioning will turn off

C: The temperature will increase

D: The temperature will decrease

**Video Source: Self-recording**

*Single-choice*

**Answer:** B

Figure 20: Visualization of samples in **State Change Prediction (Future)**.

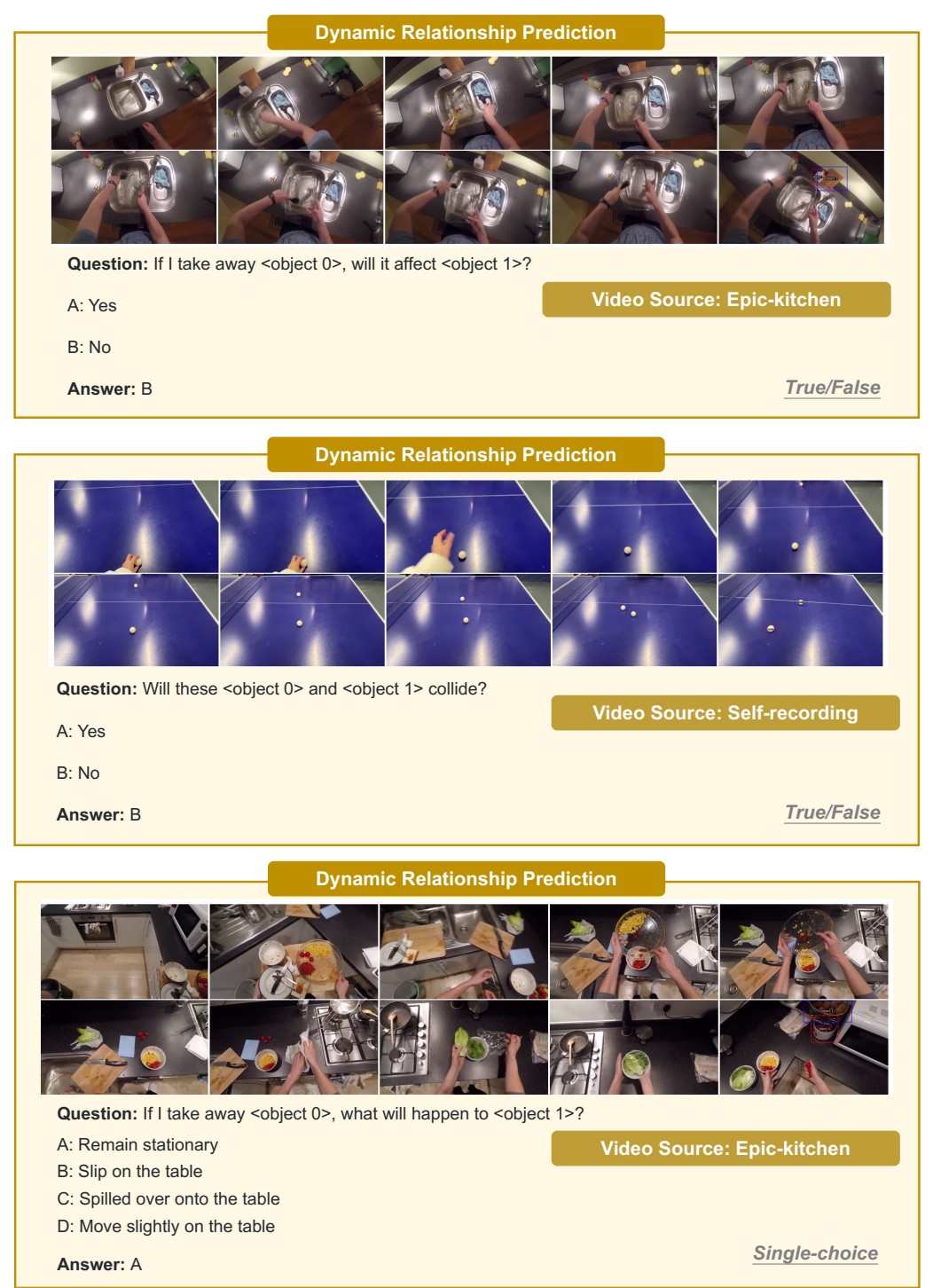

Figure 21: Visualization of samples in **Dynamic Relationship Prediction (Future)**.

