# OpenReview forum: "EOC-Bench: Can MLLMs Identify, Recall, and Forecast Objects in an Egocentric World?"
_NeurIPS.cc/2025/Datasets_and_Benchmarks_Track — NeurIPS 2025 Datasets and Benchmarks Track poster_

### Official Review · Reviewer_oqbh · 2025-06-27

**Rating:** 4
**Confidence:** 2

**Summary:**

This paper introduces EOC-Bench, a novel benchmark for evaluating multimodal large language models (MLLMs) in dynamic egocentric scenarios. The core goal is to assess object-centric embodied cognition across three temporal dimensions: Past (e.g., object state retrospection), Present (e.g., anomaly perception), and Future (e.g., trajectory prediction). The benchmark includes 3,277 QA pairs from 656 videos, leveraging visual prompts (points, boxes, masks) to resolve object referencing ambiguities in dynamic scenes. Experiments with GPT-4o, InternVL2.5, and other models reveal significant challenges in temporal reasoning (e.g., absolute time perception) and multi-frame processing, highlighting EOC-Bench’s utility for advancing MLLMs in robotics and AR/VR.

**Dataset Code Accessibility:**

Yes

**Ethical Considerations:**

No, there are no or only very minor ethics concerns

**Final Justification:**

The authors have responded to the key concern regarding dataset scale and diversity with clear plans for future work, including the incorporation of outdoor environments, multi-agent interactions, and the development of large-scale task-specific datasets with fine-tuning baselines. While the current work retains limitations in dataset diversity, the authors' commitment to addressing these gaps in future iterations is reassuring. Given their constructive response and the potential of EOC-Bench as a foundational resource, the original rating of "borderline accept" remains appropriate.

**Limitations Weaknesses:**

- Lack of Dataset Scale: With 656 videos, EOC-Bench is smaller than some prior benchmarks (e.g., MVBench: 3,673 videos), potentially limiting its generalizability to rare or long-tail scenarios.
- Lack of Diversity: Self-recorded videos focus on specific domains (e.g., electrical appliances), lacking coverage of outdoor or multi-agent interactions.
- Temporal Reasoning Depth: While the Past/Future categories are innovative, the evaluation of causal reasoning (e.g., object interactions over time) is limited. For example, "Dynamic Relationship Prediction" focuses on simple collisions rather than complex physical interactions.
- Model Evaluation Scope: Only zero-shot inference is tested, excluding fine-tuning scenarios that could reveal models’ adaptability to embodied tasks.

**Strengths Contributions:**

- Innovative Benchmark Design: Addresses a critical gap in existing benchmarks by focusing on dynamic object interactions and temporal reasoning (Past/Present/Future), which are essential for embodied AI but underrepresented in prior work (e.g., MVBench, OpenEQA). Introduces visual prompts (point/box/mask) to resolve ambiguities in egocentric scenes, a novel approach that enhances object referencing accuracy.
- Comprehensive Evaluation: Tests a diverse set of models (proprietary, open-source, object-level) under zero-shot conditions, offering a holistic view of MLLMs’ capabilities. Highlights key limitations in current models, such as poor performance in absolute time perception (open-source models score only ~15.54%) and multi-frame reasoning (open-source gains <5% with 32 frames vs. 24.6% for GPT-4o), providing clear directions for model improvement.
- High-Quality Data and Reproducibility: Integrates multiple public datasets (EPIC-KITCHENS, Ego4D) and self-recorded videos, covering diverse scenarios (kitchen, industrial, anomaly perception). Provides detailed annotation protocols and plans to open-source data/code, ensuring reproducibility and community adoption.

---

> ### Author Rebuttal · Authors · 2025-07-30
>
> We sincerely thank the reviewer for the detailed and thoughtful evaluation. Your feedback affirms the contributions in advance object-centric embodied cognition in egocentric environments.  Below, we provide point-by-point responses to each of your concerns.
>
> **Q1: About the lack of dataset scale.**
>
> **R:** Thank you for the comment. While EOC-Bench contains 656 egocentric videos, which is fewer than some larger-scale benchmarks such as MVBench, we emphasize depth and annotation diversity over raw video count.  Specifically:
> - EOC-Bench is meticulously curated with **3,277 fine-grained QA pairs**, spanning different timestamps and diverse objects. These questions cover **11 cognitively distinct tasks** across Past, Present, and Future dimensions, enabling targeted evaluation of object-centric reasoning rather than relying solely on dataset size.
> - Our benchmark prioritizes **object-level referring**, **temporal reasoning**, and **multi-modal prompts** (point/box/mask), which are often missing or sparsely represented in existing larger-scale benchmarks.
> - To mitigate generalization concerns, we  have carefully ensure that  each cognitive task includes **diverse object types**, **temporal patterns**, and **scene contexts**. Furthermore, we use **self-recorded videos** to capture both common scenarios and long-tail scenarios (e.g., anomaly events, physical interactions) that are difficult to obtain from public datasets.
>
> We believe EOC-Bench serves as a complementary and diagnostic benchmark that fills important gaps in existing benchmarks. Looking ahead,  we are actively collecting  additional egocentric videos to further expand the benchmark’s scale and task diversity in future iterations.
>
>
> **Q2: About the lack of diversity in self-recorded videos.**
>
> **R:** Our self-recorded videos are designed to **complement existing public datasets**  included in EOC-Bench, which already cover a wide range of scenarios (e.g., Ego4D, EPIC-KITCHENS, Charades-Ego and MECCANO). We primarily focus on indoor environments (e.g., kitchens, living rooms), which are highly relevant to **real-world** embodied applications such as **tool usage,  household manipulation**. These scenes exhibit dense object interactions, frequent occlusions, and complex temporal changes, making them challenging and valuable for cognition evaluation.
>
> We agree that additional diversity would be beneficial. In future iterations, we plan to **incorporate outdoor environments and multi-agent interactions** to further enhance the dataset’s representativeness and applicability.
>
> **Q3: About the temporal reasoning depth.**
>
> **R:** Thank you for highlighting this point. The Dynamic Relationship Prediction is one of three future-oriented tasks, alongside State Change Prediction and Trajectory and Motion Prediction. These tasks assess **not just simple collisions** but also temperature changes, displacements, and multi-object interactions under evolving physical contexts.
>
> For example, the questions include:
> - “What will happen to object B if object A is removed?” (cause-effect reasoning)
> - “How will these two balls move after colliding?” (motion trajectory prediction)
> - “What change will occur to the object A (cup) after being filled with object B (hot water)?” (thermal dynamics)
>
> While we acknowledge that deeper causal modeling is still an open challenge, our benchmark provides a scalable foundation upon which more complex interactions (e.g., force dynamics, affordances) can be integrated in future iterations.
>
> We will carefully clarify this in the revision. Thank you once again for your thoughtful insights.
>
> **Q4: Lack of fine-tuning scenarios in model evaluation.**
>
> **R:** Thank you for your valuable suggestions.
> Due to the lack of publicly dynamic egocentric datasets with fine-grained object annotations and temporal reasoning tasks, it remains challenging to fine-tune models for such scenarios.  In our work, the zero-shot inference  evaluation aligns with the current trend in evaluating general-purpose MLLMs and highlights their out-of-the-box capabilities in embodied cognition.
>
> To support fine-tuning scenarios, we are actively constructing large-scale, task-specific datasets and exploring fine-tuning baselines as part of our future work. We believe that EOC-Bench can inspire further research in this area and serves as a strong foundation for evaluating both generalization and adaptability in object-centric embodied cognition.

---

> > ### Author Response · Authors · 2025-08-05
> > **Looking Forward to Your Feedback**
> >
> > Dear Reviewer oqbh,
> >
> > We sincerely thank you again for your thoughtful comments on our submission.  We have provided detailed responses in the rebuttal and would greatly appreciate it if you could kindly take a moment to review our clarifications.
> >
> > If you have any further suggestions or concerns, we would be more than happy to address them. Thank you again for your time and efforts.
> >
> > Best regards,
> >
> > Authors

---

> > > ### Comment · Reviewer_oqbh · 2025-08-05
> > > **Official Comment**
> > >
> > > The authors have responded to the key concern regarding dataset scale and diversity with clear plans for future work, including the incorporation of outdoor environments, multi-agent interactions, and the development of large-scale task-specific datasets with fine-tuning baselines. While the current work retains limitations in dataset diversity, the authors' commitment to addressing these gaps in future iterations is reassuring. Given their constructive response and the potential of EOC-Bench as a foundational resource, the original rating of "borderline accept" remains appropriate.

---

> > > > ### Author Response · Authors · 2025-08-06
> > > >
> > > > Thank you very much for your thoughtful review and feedback. We sincerely appreciate your positive acknowledgment of EOC-Bench's potential as a foundational resource, and we are grateful for your recognition of our detailed future plans for expansion.
> > > >
> > > > We would like to kindly highlight that, in the current form, EOC-Bench uniquely addresses critical gaps in existing egocentric datasets through meticulous fine-grained object annotations and cognitively rich temporal reasoning tasks across Past, Present, and Future dimensions. Its cognitive scope has been  broadened by combining rich and diverse public datasets, such as Ego4D, EPIC-KITCHENS, Charades-Ego, and MECCANO, with self-recorded long-tail scenarios. This design fills an important gap in the evaluation of object-centric embodied cognition in egocentric settings.
> > > >
> > > > We believe that the current contributions offer significant diagnostic and complementary value to the community, merit recognition as a strong and impactful resource in this emerging research direction, while our ongoing expansions will further enhance its scope.
> > > >
> > > > Thank you once again for your thoughtful review.  We deeply appreciate your time and effort for evaluating our work.

---

### Official Review · Reviewer_5hRs · 2025-06-30

**Rating:** 5
**Confidence:** 4

**Summary:**

This paper introduces EOC-Bench, a novel benchmark for evaluating the embodied object cognition capabilities of Multimodal Large Language Models (MLLMs) in egocentric, dynamic, and temporally evolving environments,filling the gap of existing research that focuses on static scenes and spatial attributes while ignoring dynamic changes caused by user interactions. The benchmark contains 3,277 annotated QA pairs categorized into three dimensions: past, present, and future, with human-in-the-loop annotation and visual prompts such as bbox, point, mask. Extensive evaluations on various proprietary and open-source MLMs are conducted, and limitations in terms of object-level temporal understanding are highlighted.

**Dataset Code Accessibility:**

Yes

**Dataset Code Comments:**

The authors provide a Markdown file in their supplementary to guide the use of the dataset and code, environment configuration is also provided, and make publicly available the test code for the MLLMs used in the paper.

**Ethical Considerations:**

No, there are no or only very minor ethics concerns

**Final Justification:**

The author has addressed my concern, and I would like to maintain my rating as "Accept".

**Limitations Weaknesses:**

1. While the benchmark covers 11 object cognition tasks across Past, Present, and Future dimensions, the paper lacks an explicit breakdown of how the 3,277 QA pairs are distributed among the 11 categories. For example, from Section 3.3 we know that only 507 questions belong to the “Future” dimension, but it remains unclear how many samples are assigned to each subtask. Uneven category distribution may introduce bias during model evaluation and training. A more detailed statistical breakdown would help assess whether the benchmark adequately represents all cognition skills.

2. The paper needs more analysis about the experiemens. e.g., it lacks a deeper analysis of the underlying causes. It remains unclear whether these failures are primarily due to model architecture limitations (e.g., insufficient temporal attention span), training data bias (e.g., insufficient coverage of historical dynamics), or deficiencies in visual prompt representations. A more granular causal breakdown would enhance the paper’s diagnostic value.

**Strengths Contributions:**

1. The proposed benchmark addresses a clear gap in existing embodied vision benchmarks—namely, the lack of temporal understanding and object-centric evaluations in dynamic, egocentric environments.

2. The benchmark includes fine-grained temporal categories, various question types (True/False, single/multiple-choice, open-ended), and visual object referencing formats. This design reflects practical embodied cognition demands.

3. A hybrid-format human-in-the-loop annotation framework comprising four question types has been developed, along with a novel Multi-Scale Temporal Accuracy (MSTA) metric for open-ended temporal evaluation. These designs enable EOC-Bench to comprehensively and accurately assess the temporal perception capabilities of MLLMs.

---

> ### Author Rebuttal · Authors · 2025-07-30
>
> We sincerely thank the reviewer for the thoughtful and constructive feedback. Your comments affirm the value of our benchmark in advancing object-centric embodied cognition in dynamic, egocentric environments. We have carefully addressed each of your points in our detailed responses below and have incorporated the corresponding suggestions to further strengthen our work.
>
> **Q1: A more detailed statistical breakdown of each subtask.**
>
> **R:**  Thanks  for  your insightful comments.  Below we provide the distribution of QA samples across the 11 object cognition tasks in EOC-Bench:
>
> | Task Category                    | # QA Pairs |
> |----------------------------------|-------|
> | Object State Retrospection       | 361   |
> | Object Location Retrospection     | 369   |
> | Object Relationship Evolution    | 359   |
> | Absolute Time Perception         | 333   |
> | Immediate State Recognition      | 597   |
> | Object Relationship              | 339   |
> | Purpose and Function Inference   | 310   |
> | Anomaly Perception               | 102   |
> | Trajectory and Motion Prediction | 205   |
> | State Change Prediction          | 214   |
> | Dynamic Relationship Prediction  | 88    |
>
> This distribution reflects the natural frequency and complexity of different cognitive tasks observed in egocentric environments—e.g., "Immediate State Recognition" is more frequent in real-world scenarios, while tasks like "Dynamic Relationship Prediction" and "Anomaly Perception"  involve higher-level reasoning under rare or ambiguous contexts, resulting in fewer but more challenging instances.
>
> While the distribution is inherently imbalanced, we have carefully ensured that **each task includes diverse and representative samples**, encompassing a wide range of temporal dynamics, object types, and visual settings.
> We will incorporate this detailed breakdown into the revised version to improve clarity.
>
> **Q2:  A more granular causal breakdown of experimental results.**
>
> Thank you for the valuable suggestion. In Appendix B.3, we provide a quantitative error analysis.  Below, we further conduct a more in-depth causal breakdown for diagnostic insights. We summarize key failure patterns observed across the three temporal categories:
>
> - **Past Category**
>   As shown in Figure 3 of the Appendix, memory-related errors account for **93%** of failures in this category, which we further classify as follows:
>
>   1. **Insufficient processing of historical frames (73%)**: The models fail to retrieve earlier object states, especially when visual evidence appears many frames before the queried moment.
>
>   2. **Interference from current frame (17%)**: Attention drifts to present context and overwrites relevant past memories.
>
>   3. **Missing observation (10%)**: Coarse or fixed-interval frame sampling skips key visual events.
>
>   **The first two types** indicate a **limited temporal attention span**, which could stem from design shortcomings in long video understanding or biases in the training dataset. Specifically, these biases may result from insufficient exposure to long-range temporal dynamics, thereby impairing the model’s ability to develop strong long-range temporal dependencies.
>   **The third type** reflects weaknesses in **frame sampling strategies**, as most MLLMs adopt uniform sampling (e.g., InternVL2.5, LLaVA-Video, LongVU), which may skip key contextual cues.
>
> - **Present Category**
>   In Present tasks, a large portion of failures arise from:
>
>   1. **Current visual perception error (28%)**: The model struggles to resolve references to visual prompts.
>   2. **Common sense error (20%)**: Misunderstandings regarding object function or usage.
>   3. **Intra-frame confusion (18%)**: Visually similar objects are misidentified.
>
>   These issues likely stem from training bias, as most MLLMs are trained without explicit visual prompts, resulting in dispersed attention and weak fine-grained spatial focus. These findings indicate that **precise spatial perception remains a persistent challenge**.
>
> - **Future Category**
>   Approximately **59%** of errors in the Future category are due to knowledge-related issues, such as incorrect predictions about object trajectories or interactions. These reveal gaps in common-sense reasoning and anticipatory inference, suggesting that current MLLMs **lack robust world-modeling capabilities**, which are crucial for forecasting in egocentric and temporally-evolving environments.
>
>
> We will incorporate this extended analysis in the revision to improve the diagnostic depth of our evaluation. Thank you once again for your thoughtful insights.

---

> > ### Comment · Reviewer_5hRs · 2025-08-02
> >
> > I would like to thank the authors for their detailed response, which has addressed my concern. I would like to maintain my rating as "Accept".

---

> > > ### Author Response · Authors · 2025-08-02
> > >
> > > Thank you very much for your timely feedback and support. We are grateful that our response was able to address your concern, and we sincerely appreciate your thoughtful review in further improving our work.

---

### Official Review · Reviewer_i3fF · 2025-07-01

**Rating:** 6
**Confidence:** 4

**Summary:**

EOC-Bench, a new benchmark designed to evaluate the object-centric embodied cognitive abilities of MLLMs in dynamic, egocentric scenarios. The authors identify a gap in existing benchmarks, which primarily focus on static scene exploration while neglecting the dynamic object interactions common in first-person operational tasks.

**Additional Feedback:**

What happens if you include RL-Based Reasoning? Don’t just perform zero-shot inference

**Dataset Code Accessibility:**

Yes

**Dataset Code Comments:**

The authors provided the code, guidelines for reproducibility, and the Huggingface link for the dataset.

**Ethical Considerations:**

No, there are no or only very minor ethics concerns

**Final Justification:**

Most of my original concerns regarding domain imbalance, analysis by video duration, and distinctions from other benchmarks have been addressed in the rebuttal. The authors provided clear justifications and supportive experiments, which satisfactorily resolve these points.

**Limitations Weaknesses:**

- There is no domain balance, Kitchen (or indoor) scenes dominate (Fig. 3.1), and outdoor contexts are scarce.

- A deeper analysis of model performance based on video duration would strengthen the experiments. For example, subcategorizing videos into short, medium, and long durations (as in MM-EGO [1]) would provide clearer insights into the models' capabilities in terms of memory and future prediction.

- I want to see a deeper analysis of different benchmarks in Table 1.  For example, EgoPlan-Bench [2] has a next action prediction capability as well. It subcategorizes based on the task-progress, current observation and the next action prediction. I would like the authors to highlight more distinctions of the EOC-Bench compared to the others.

[1] Ye, Hanrong, et al. "MMEgo: Towards Building Egocentric Multimodal LLMs for Video QA." The Thirteenth International Conference on Learning Representations, 2025, https://openreview.net/forum?id=67sSPPAZiG.

[2] Chen, Yi, et al. "Egoplan-bench: Benchmarking multimodal large language models for human-level planning." arXiv preprint arXiv:2312.06722 (2023).

**Strengths Contributions:**

-  It is the first benchmark to jointly evaluate an MLLM's ability to identify, recall, and predict object states across time, addressing a clear weakness in prior benchmarks that focused on static scenes (I couldn’t find any benchmark that jointly probes identification, recall, and prediction in egocentric video).

- The benchmark is thoughtfully designed with a Past-Present-Future framework covering 11 cognitive tasks. It also provides visual prompts for unambiguous object referencing.

- The paper is extremely well-written and clear. I like the presentation of the paper. Figures are very informative. Also, the analysis and design of experiments are interesting. The authors did a great job in benchmarking 27 MLLMs. Extensive experiments on numerous MLLMs reveal performance gaps, especially in temporal reasoning, proving EOC-Bench is a valuable tool for highlighting the limitations of current models and guiding future research.

---

> ### Author Rebuttal · Authors · 2025-07-30
>
> We appreciate that the reviewer recognizes the originality and contributions of our work. We sincerely thank the reviewer for the detailed feedback and constructive suggestions. Here, we treasure the opportunity to address your concerns and improve the quality of our work.
>
> **Q1: About domain imbalance - indoor scene dominate while outdoor contexts are scarce.**
>
> **R:** Thank you for the thoughtful comment. We fully agree that a more balanced coverage across indoor and outdoor scenarios would benefit generalization of benchmark.
> The current distribution of EOC-Bench primarily consists of indoor environments (e.g., kitchens, living rooms), as these settings are most representative of real-world egocentric embodied applications such as tool usage and household object manipulation. These scenes present rich and cluttered object interactions, frequent occlusions, and complex spatial-temporal dynamics, making them particularly challenging and valuable for assessing object-centric cognitive abilities. In the future iterations of EOC-Bench, we will incorporate a broader range of outdoor and diverse contexts to further expand its coverage and applicability.
>
>
> **Q2: Analysis of model performance based on video duration.**
>
> **R:** We appreciate your insightful suggestion regarding a deeper analysis of model performance across different video durations. We have conducted a detailed evaluation by subcategorizing videos into short, medium, and long durations, like MM-EGO [1] as you recommended.  Below is a summary of the model performance across these duration categories:
>
>
> | Model                | Input | Mean (Short) | Mean (Medium) | Mean (Long) | Past (Short) | Past (Medium) | Past (Long) | Present (Short) | Present (Medium) | Present (Long) | Future (Short) | Future (Medium) | Future (Long) |
> |----------------------|-------|--------------|---------------|-------------|--------------|---------------|-------------|-----------------|------------------|----------------|----------------|-----------------|---------------|
> | GPT-4o            | 32f   | 62.57 | 61.33 | 58.36 | 57.40 | 52.74 | 49.41 | 66.36   | 69.39 | 75.47 | 65.52  | 75.80 | 68.97 |
> | Gemini-2.0-flash  | 32f   | 57.75 | 56.61 | 57.07 | 50.09 | 43.11 | 48.72 | 65.33   | 67.70 | 67.05 | 53.77  | 71.34 | 67.24 |
> | GPT-4o-mini       | 32f   | 50.17 | 48.25 | 48.21 | 40.10 | 37.07 | 42.19 | 58.51   | 58.08 | 56.82 | 50.34  | 59.24 | 53.45 |
> | InternVL2.5-78B   | 32f   | 52.60 | 52.52 | 50.15 | 44.17 | 37.29 | 38.19 | 60.37   | 65.29 | 64.77 | 50.34  | 70.06 | 65.52 |
> | InternVL2.5-38B   | 32f   | 53.32 | 51.37 | 48.40 | 43.44 | 35.41 | 34.48 | 61.40   | 68.38 | 68.18 | 54.11  | 63.06 | 62.07 |
> | Qwen2.5-VL-72B    | 1fps  | 50.40 | 49.03 | 48.78 | 39.68 | 37.18 | 38.11 | 58.72   | 59.45 | 61.23 | 54.44  | 61.78 | 59.62 |
> | LLaVA-Video-72B   | 32f   | 49.81 | 47.39 | 47.71 | 41.10 | 33.82 | 34.89 | 56.55   | 62.20 | 65.91 | 51.71  | 56.69 | 60.34 |
> | LLaVA-OV-72B      | 32f   | 48.39 | 46.48 | 48.40 | 36.01 | 32.41 | 35.03 | 58.20   | 63.57 | 65.91 | 50.34  | 52.87 | 63.79 |
> | VideoLLaMA3-7B    | 1fps  | 46.41 | 47.31 | 40.32 | 36.99 | 32.47 | 32.01 | 53.87   | 63.92 | 53.41 | 47.96  | 56.69 | 46.55 |
> | InternVL2.5-8B    | 32f   | 42.21 | 41.26 | 43.17 | 33.57 | 32.44 | 35.23 | 49.54   | 49.83 | 51.14 | 41.78  | 48.41 | 55.17 |
> | VideoLLaMA2.1-72B | 16f   | 41.85 | 41.35 | 40.17 | 33.10 | 31.53 | 28.43 | 46.85   | 51.55 | 54.55 | 49.66  | 49.04 | 55.17 |
> | LongVA-7B         | 32f   | 36.46 | 32.50 | 35.75 | 29.08 | 27.94 | 28.71 | 41.59   | 35.40 | 43.18 | 40.07  | 39.49 | 46.55 |
>
> Across all models, we observe a consistent performance drop as video length increases. For instance, GPT-4o achieves a mean accuracy of 62.57 on short videos, compared to 61.33 on medium and 58.36 on long ones. Other models  exhibit similar trends. This trend highlights that most MLLMs demonstrate stronger capabilities in short-term perception and recall, but struggle to maintain memory consistency over extended temporal contexts.
>
> The most significant performance degradation occurs in the **Past** category, indicating that long-term memory recall remains a critical challenge for current MLLMs in egocentric scenarios. In contrast, **Present** and **Future** tasks tend to benefit from medium or long video durations, indicating that extended temporal context helps improving scene understanding and anticipatory reasoning.
>
> We will include these analysis in the revision to provide deeper insight into models' memory retention and future prediction capabilities.
>
>
> **Q3: A deeper analysis of distinctions compared to other benchmarks in Table 1.**
>
> **R:** Thanks for your thoughtful suggestion. EgoPlan-Bench [2] focuses on high-level task planning and next-action prediction. In contrast, EOC-Bench is centered on **object-centric embodied cognition**, evaluating a model’s ability to **identify (Present), recall (Past), and forecast (Future)** specific objects over time.
>
> Key differences include:
>
> - **Temporal Scope:** EgoPlan-Bench mainly focuses on the present-to-future transition, while our benchmark spans past, present, and future cognition.
> - **Cognitive Focus:** EOC-Bench centers on object-level understanding, while EgoPlan-Bench targets human activity prediction and task progress.
> - **Fine-grained Visual Understanding:** EOC-Bench uniquely supports point, box, and mask prompts, enabling fine-grained, localized object reasoning, which is not included in EgoPlan-Bench.
>
> We will make these distinctions clearer in the revision and extend Table 1 and the discussion accordingly.
>
>
> **Q4: Regarding RL-based reasoning. (Additional feedback)**
>
> **R:** Thank you for the insightful comment. In this work, we focus on the zero-shot evaluation setting to assess the inherent object-level reasoning capabilities of pretrained MLLMs. We agree that incorporating RL-based training could further enhance the performance, especially in Past and Future tasks. Exploring such approaches is an exciting direction for future work, and we plan to construct task-specific datasets to support RL-based training to  strengthen models’ object-centric reasoning in the ego-centric world.
>
> **References**
>
> [1] Ye, Hanrong, et al. "MMEgo: Towards Building Egocentric Multimodal LLMs for Video QA." The Thirteenth International Conference on Learning Representations, 2025.
>
> [2] Chen, Yi, et al. "Egoplan-bench: Benchmarking multimodal large language models for human-level planning." arXiv preprint arXiv:2312.06722, 2023.

---

### Official Review · Reviewer_KJVD · 2025-07-03

**Rating:** 5
**Confidence:** 5

**Summary:**

The paper introduces EOC-Bench, an egocentric video understanding benchmark consisting of curated samples from four existing ego video datasets and another self-recorded datasets. The main contribution of the benchmark is diverse spatiotemporal object-centric human annotations for different types of tasks concerning objects relations, movement dynamics, absolute time perception, and functionality spanning three temporal categories (present, past, and future). The label types are either binary, single/multiple choice, or open-ended (seconds for absolute time perception). The authors evaluate multiple  proprietary and open-source MLLMs zero-shot and compare their performances with random and human baselines. They provide a discussion on the shortcomings of these models and highlight the need for advancing spatiotemporal reasoning capabilities of vision-language foundation models.

**Dataset Code Accessibility:**

Partly

**Dataset Code Comments:**

See my first point in Limitations.

**Ethical Comments:**

Please specify the monetary compensation of annotators. Please also specify the exact procedure for collecting the self-recorded videos (and if there was any monetary compensation associated with it).

There are no concerns regarding external datasets as they are open-sourced. Evaluated models are either open-weight or provide API access.

**Ethical Considerations:**

No, there are no or only very minor ethics concerns

**Final Justification:**

I can now access the video data. Since my main concern has been resolved, I am increasing my score.

**Limitations Weaknesses:**

My main concern which prevents me from giving a higher score now is the accessibility to the dataset. The only available file on hugging face is the test split annotation file. Neither the video samples from external datasets nor the self-recorded videos are available (although the evaluation repo on github is mostly clean and well-documented). I hope the authors provide the video files alongside complete annotations.

**Strengths Contributions:**

1. Human annotations and task variety in the benchmark are very rich and diverse and a systematic categorisation is performed for three temporal categories (which I have not seen in the literature for object-centric egocentric video understanding). A benchmark such as EOC is needed to evaluate the spatiotemporal reasoning capabilities of foundation models that will be increasingly deployed on embodied robotic agents in the coming years.

2. SoTA proprietary MLLMs and multiple smaller open-source models are evaluated and compared on all tasks (in terms of of task type, label type and temporal category) and their shortcomings and points of weakness are discussed in section 4. Complementing evaluation results and discussions are provided in Appendix B.

3. The text is well-written, well-organized, and easy to follow. Figures are visually appealing and informative.

4. Enough evaluation details are provided in main text and appendix for reproducibility.

---

> ### Author Rebuttal · Authors · 2025-07-30
>
> We sincerely thank you for the thoughtful and constructive feedback. We deeply appreciate your recognition of EOC-Bench's value in assessing spatiotemporal object-centric reasoning, which we fully agree is critical for advancing embodied robotic agents. We treasure the opportunity to address your concerns and improve the quality our work.
>
> **Q1: Concerning  accessibility of video samples.**
>
> **R:** Sorry for the confusion. In fact, **all videos**, including those from external datasets as well as our self-recorded videos, are fully available through our Hugging Face repository in the `videos` folder (provided as three tar files).
> The folder structure is as follows:
> ```
> ├── data
> │   └── test-00000-of-00001.parquet
> └── videos
>     ├── videos.tar.part.aa
>     ├── videos.tar.part.ab
>     └── videos.tar.part.ac
> ```
>
> To avoid any misleading, we will update the repository's README to clearly state the availability of all video files and include a clear explanation of the folder structure. We hope this clarification helps the community fully leverage EOC-Bench for future research.
>
> **Q2: Clarification regarding the monetary compensation of annotators and the procedure for collecting self-recorded videos.**
>
> **R:** All annotators were fairly compensated for their time and contributions. Each annotator was paid `$10 per hour` for their work.
> For the self-recorded videos, we recruited five volunteers.   To ensure consistency and quality, we provided detailed recording guidelines. Videos were captured using smartphones to support environmental diversity. Each volunteer was compensated `$8 per video` for their efforts.

---

> > ### Comment · Reviewer_KJVD · 2025-08-03
> >
> > Thanks for your response.
> >
> > I can now access the video data. Since my main concern has been resolved, I am increasing my score.

---

> > > ### Author Response · Authors · 2025-08-03
> > >
> > > Thank you very much for your follow-up and for confirming access to the video data. We sincerely appreciate your time and your decision to increase the score. We are truly grateful for your support.

---

### Decision · Program_Chairs · 2025-09-18

**Decision:**

Accept (poster)

**Comment:**

This paper presents a dataset for benchmarking multimodal large language models in egocentric videos. Videos are taken from 4 public egocentric video databases plus in-house recordings. The benchmark contains 3277 question-answer pairs categorized across different dimensions, e.g., temporal (past, present, future) or visual prompts (point, box, mask). The authors evaluate several baselines, highlighting strengths and weaknesses, e.g., their struggles with temporal reasoning.

The reviewers have in general a positive view of this work, highlighting the high quality, richness, diversity and structure of the data, its potential value for progress in the field and the exhaustive evaluation and interesting analysis of the results. Data imbalance was raised, which is a minor concern nicely addressed by the authors. The paper needs further analysis and statistics regarding video duration and task categories, that the authors also addressed satisfactorily in the rebuttal. Comments in this regard should be incorporated to the final version of the paper.